# Glacial melt disturbance shifts community metabolism of an Antarctic seafloor ecosystem from net autotrophy to heterotrophy

Ulrike Braeckman [1,2 ✉], Francesca Pasotti[1], Ralf Hoffmann [2], Susana Vázquez [3], Angela Wulff[4], Irene R. Schloss [5,6,7], Ulrike Falk[8], Dolores Deregibus[5,9], Nene Lefaible[1], Anders Torstensson [4], Adil Al-Handal[4], Frank Wenzhöfer [2,10] & Ann Vanreusel[1]

Climate change-induced glacial melt affects benthic ecosystems along the West Antarctic Peninsula, but current understanding of the effects on benthic primary production and respiration is limited. Here we demonstrate with a series of in situ community metabolism measurements that climate-related glacial melt disturbance shifts benthic communities from net autotrophy to heterotrophy. With little glacial melt disturbance (during cold El Niño spring 2015), clear waters enabled high benthic microalgal production, resulting in net autotrophic benthic communities. In contrast, water column turbidity caused by increased glacial melt run-off (summer 2015 and warm La Niña spring 2016) limited benthic microalgal production and turned the benthic communities net heterotrophic. Ongoing accelerations in glacial melt and run-off may steer shallow Antarctic seafloor ecosystems towards net heterotrophy, altering the metabolic balance of benthic communities and potentially impacting the carbon balance and food webs at the Antarctic seafloor.

[1] Marine Biology Research Group, Ghent University, Ghent, Belgium. [2] HGF MPG Joint Research Group for Deep-Sea Ecology and Technology, Alfred-Wegener-Institut, Helmholtz Zentrum für Polar-und Meeresforschung, Bremerhaven, Germany. [3] Cátedra de Biotecnología, Facultad de Farmacia y Bioquímica, Universidad de Buenos Aires, Nanobiotec UBA-CONICET, Ciudad Autónoma de Buenos Aires, Argentina. [4] Department of Biological and Environmental Sciences, University of Gothenburg, Gothenburg, Sweden. [5] Instituto Antártico Argentino, Buenos Aires, Argentina. [6] Centro Austral de Investigaciones Científicas, Ushuaia, Tierra del Fuego, Argentina. [7] Universidad Nacional de Tierra del Fuego, Ushuaia, Tierra del Fuego, Argentina. [8] Bremen University, Bremen, Germany. [9] Consejo Nacional de Investigaciones Científicas y Técnicas (CONICET), Ciudad Autónoma de Buenos Aires, Ciudad Autónoma de Buenos Aires, Argentina. [10] Max Planck Institute for Marine Microbiology, Bremen, Germany. ✉email: Ulrike.Braeckman@ugent.be

The West Antarctic Peninsula (WAP) has undergone rapid and significant warming during the second half of the 20th century[1]. The sea ice season has shortened by about 100 days[2,3] and 87% of coastal glaciers are in retreat[4,5]. These alterations in the cryosphere have strong consequences for marine ecosystems[6], but the effects on the metabolic balance of the benthic communities are poorly quantified.

When marine-terminating glaciers calve, the new bare space available at the seafloor can be colonized by, e.g. benthic microalgae[7], macroalgae[8,9], benthic filter feeders on hard substrates[10] and new soft substrate communities[11]. However, marine-terminating glaciers may also support high productivity through rising subsurface meltwater plumes with nutrient-rich deep water[12,13]. This effect disappears when these glaciers retreat on land[12]. The increased glacial melting and related ice scour frequency[14] may also lead to higher ice scouring-induced mortality[15–17], stress for filter feeders by sediment resuspension[18], and consequently a reshaping of the benthic assemblages[11,19]. As a result, more frequent ice scouring in coastal WAP areas may alter patterns in benthic community respiration (CR) and decomposition of dead material in the ice-scoured tracks.

During the melt season sub-glacial melt induces yet another factor influencing the carbon balance in shallow systems, when an up to 5 m-thick turbid water column layer develops. Lithogenic particles from coastal- and subglacial run-off can then restrict phytoplankton[20], macroalgal[21] and benthic microalgal production[22] through attenuation of underwater radiation. In shallow high latitude systems where benthic microalgae are the main primary producers, a reduction in photosynthetically active radiation availability due to increased turbidity in combination with increased sedimentation[22] can determine a shift in benthic metabolism from an autotrophic to a heterotrophic state[23].

This shift can have important consequences for biogeochemical cycling, bentho-pelagic coupling and ultimately, food webs on the WAP.

Finally, the WAP is also subject to strong interannual atmospheric climate variability, driven primarily by the Southern Annular Mode (SAM; in positive phase: +SAM, in negative phase: −SAM)[24,25] and El Niño Southern Oscillation (ENSO: El Niño–La Niña conditions). These long-term atmospheric oscillations affect winds and air temperature. In the northern WAP, +SAM conditions increase the westerlies, whereas La Niña conditions increase northerly winds; both processes synergistically leading to an increased heat influx to the WAP[26]. Combined +SAM and La Niña conditions since the late 1980s onwards have been shown to delay autumn sea ice advance and shorten the sea ice season (earlier sea ice retreat)[3,25], possibly acting as an additional, indirect forcing factor on benthic metabolism.

Here, we investigated benthic community metabolism based on net production and respiration at three sites with differing glacial melt impact in Potter Cove, King George Island/Isla 25 de Mayo (Fig. 1, Supplementary Table 1), during summer, winter and spring 2015 (strong El Niño year) and in spring 2016 (weak La Niña year) and examined environmental drivers in climatology, cryosphere and oceanography that relate to glacial melt conditions. We hypothesized that benthic autotrophy is only possible in conditions with little glacial melt disturbance. In addition, by analysing the spatial and temporal trends in our data, we provide an insight into the benthic carbon balance in future climate change scenarios for coastal WAP areas.

## Results and discussion

**Climatology, cryosphere and oceanography.** Strong winter El Niño conditions were observed in 2015, indicated by a more positive ENSO MEI index (average 2015: 1.3). In early 2016, the climatic conditions reversed, turning into weak La Niña conditions with a lower index (average 2016: 0.5) (Fig. 2a). Throughout the study period, the SAM index was mainly in a positive phase. Interannual climate variability by SAM acts on decadal scales[27,28], while ENSO cycles occur on sub-decadal scales and are therefore more relevant for short-term interannual variability. Northerly winds bring relatively warm air to the northern WAP, but during an El Niño year such as 2015, northerly winds weaken, and cold air reaches the area, resulting in extended sea ice cover periods (earlier advance and later retreat)[25] and slower glacial melt rates[29,30]. The El Niño winter of 2015 was indeed ~3 °C colder than the La Niña winter 2016 (Fig. 2b), and was characterized by a long sea ice cover period (118 days—nearly 4 months; Fig. 2c) and a late onset of sea ice and glacial melt (Fig. 2d). Between 1989 and 2015 there were on average $104 \pm 39$ days of sea ice[31]. A long sea ice cover period such as that of 2015 had been observed before 2003 and then again only in 2011[31]. Generally, there is hardly any glacial meltwater run-off in winter, with the onset of run-off usually happening in spring (October), peaking in summer (January–March) and returning to zero in early winter (June–July)[30]. The maximum estimated run-off during the period 2011–2016 was about 3 m³ s$^{-1}$[30]. No glacial melt run-off was observed during winter 2015, and the onset of run-off was delayed with one month to November (Fig. 2d). As such, cold air temperatures during early spring 2015 prevented glacier melting and high particle-laden freshwater to enter Potter Cove waters until later that summer (Fig. 2h: suspended particulate matter (SPM) concentrations in winter and spring 2015 were half those of summer 2015 and spring 2016). These conditions resulted in remarkably clear waters, which led to high chlorophyll-a (chl-a) concentrations in the water column in spring 2015 (Fig. 2i). The prolonged peak from December 2015 to March 2016 is likely related to enhanced phytoplankton production due to a combination of optimal stratification and light conditions, as previously observed[32]. Such a consistent positive response of phytoplankton to more persistent winter sea ice during El Niño and/or −SAM events is well known for the northern WAP[20,33]. At the end of an El Niño year, northerly winds strengthen, bringing warmer air to the region[25,34]. The peak in glacial run-off during austral summer 2015/2016 occurred in February 2016, but maximum run-off was ~10 m³ s$^{-1}$ (Fig. 2d), about three times higher than in the years 2010–2015[30]. This fast, increased run-off may have resulted from the sustained high air temperatures that accompanied the shift between El Niño to La Niña conditions[25]. Although 2016 was a weak La Niña year, its effect was reinforced by +SAM conditions[35]. Higher air temperatures also contribute to postponing sea ice advance and accelerating sea ice retreat, hence resulting in shorter sea ice periods[25]. Sea ice cover periods during the warmer La Niña winter of 2016 were indeed short and interrupted (2 weeks mid-winter and 1.5 months end of winter; Fig. 2c). Sea ice has an important feedback on glacial melt run-off, since it functions as a barrier to the meltwater discharge in the cove. Consequently, a substantial glacial run-off was observed in winter 2016, equal in magnitude to the summer run-off estimates in 2015 (Fig. 2d). In addition, there had not been any net snow accumulation on the Fourcade glacier during winter 2016 (U. Falk, personal communication). Without snow accumulation during winter, the dark material from rocks in bare-ice conditions is known to reinforce surface melt, with higher rates of glacier discharge during the following summer[30]. Indeed, the subsequent austral summer of 2016–2017 was characterized by exceptionally high (max. in February 2017 ~14 m³ s$^{-1}$, about 4.5 times higher than the 2010–2015 maxima) glacier discharge rates (Fig. 2d), resulting in a >2 times larger SPM peak in the water column in spring 2016 compared to spring 2015 (Figs. 2h and 5).

Also distinct lows in sea surface salinity were observed, pointing at melt water input in summer 2016 (Fig. 2g).

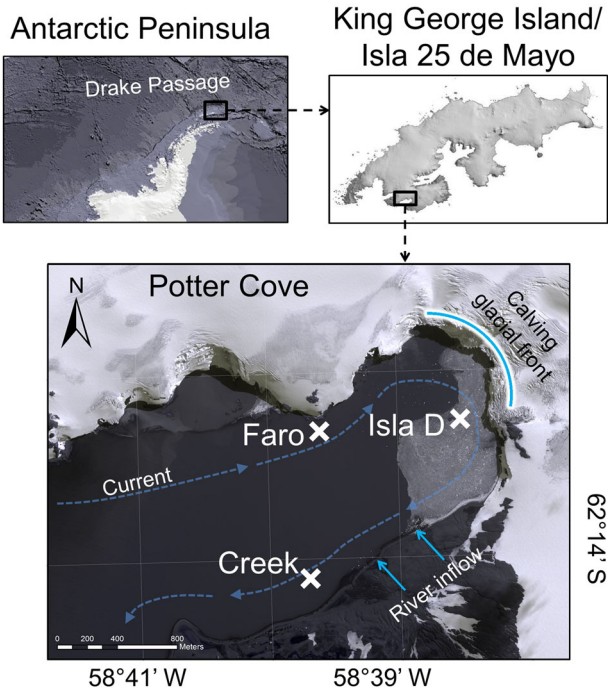

**Fig. 1 Study site.** In situ measurements and sediment sampling were conducted at three sites—Faro, Creek, and Isla D (positions are marked with a cross). The curved, bright blue line marks the front of the Fourcade glacier. The bright blue arrows indicate river run-offs from the creeks, supplied mainly by melting glacier, permafrost and snow. The dashed blue arrows indicate the direction of the main current in Potter Cove. Owing to this current direction, Faro is considered an upstream site experiencing little glacial melt disturbance, Isla D is a glacier front site under high glacial melt influence and Creek is located downstream of a melt water river, experiencing glacial melt disturbance in addition to meltwater plume influence. Satellite picture from Potter Cove taken on 3 November 2013 from Digital Globe[84].

Consequently, and due to the very low wind speeds (Fig. 2e), the suspended sediment particles stayed in the cove. This increased turbidity shaded the water column, likely leading to more interrupted pelagic phytoplankton productivity[36] in spring 2016 (Fig. 2i). The high, short lived chl-*a* values detected in January 2017 were probably advected from neighbouring ice-free and particle-free areas by the dominating westerly winds. This allowed phytoplankton to grow during a short period (3 days) of relatively calm winds, but they were swept away when wind intensity increased again, as on January 16, 2017 (Fig. 2i).

**Benthic community metabolism.** The net community metabolism (NCM; total oxygen exchange (TOE) rate measured in transparent benthic chambers deployed at the sea floor) ranged between −42.8 and 18.9 mmol $O_2$ m$^{-2}$ d$^{-1}$ (Fig. 3a). Net primary production by benthic microalgae (positive NCM) was mainly observed in spring 2015, when photosynthetic active radiation (PAR) measured at the seafloor clearly surpassed 26 µmol m$^{-2}$ s$^{-1}$, the experimentally determined light compensation point at Faro and Isla D[22] (no PAR data for Creek) (Supplementary Fig. 1; Supplementary Dataset 4). Apart from some patches at the least glacial-disturbed site Faro in spring 2016, there was no net benthic microalgal production in spring 2016 or winter and summer 2015 (see some replicates with net production (positive NCM) in Fig. 3a). This absence of net benthic production corresponds with PAR lower than the light compensation point[22] in summer (Faro, Isla D) and winter 2015 (Faro), but not in spring 2016, when

measured PAR at each site surpassed the light compensation point[22] (Supplementary Fig. 1). As discussed in Hoffmann et al. [22], the lower to absent benthic microalgal production in spring 2016 is probably not attributed to unavailability of PAR at the sea floor, but rather to direct physical disturbance of the benthic microalgae: damage of the benthic diatom photosynthetic apparatus can occur through sediment accumulation[37]. Sedimentation also affects benthic microalgae through the longer distances that diatoms have to migrate to find the best light conditions[38], which affects energy allocation, thereby lowering the overall net production[22]. With similar CR rates as in 2015, ranging between −9.6 and −43.8 mmol $O_2$ m$^{-2}$ d$^{-1}$ (Fig. 3b), the calculated gross primary production (GPP) ranged between −3.3 and 59.8 mmol $O_2$ m$^{-2}$ d$^{-1}$ and was highest in spring 2015 (41–60 mmol $O_2$ m$^{-2}$ d$^{-1}$), followed by spring 2016 (1–26 mmol $O_2$ m$^{-2}$ d$^{-1}$) and close to zero in summer (−3–1 mmol $O_2$ m$^{-2}$ d$^{-1}$) and winter 2015 (−2 to 2 mmol $O_2$ m$^{-2}$ d$^{-1}$) (Fig. 3c). Taken altogether, the carbon balance in Potter Cove sediments turned from net autotrophy (GPP > CR) in spring 2015 to net heterotrophy (GPP < CR) in spring 2016, especially in the turbid area close to the glacier (Isla D) and in the meltwater plume area (Creek).

**Biotic and environmental drivers of benthic community metabolism.** A strong spatiotemporal gradient in benthic community parameters was evident between winter and spring 2015 and the other two seasons, as well as between sites far from the glacier (Faro and Creek) and closer to the glacier front (Isla D) (Fig. 4; Comp.1). This gradient explained 27% of the variation. High benthic net primary production (positive NCM; Fig. 3a) and biomass in terms of benthic microalgae (Fig. 3e) and prokaryotes (Supplementary Fig. 2; Supplementary Dataset 5), as well as high chl-*a* (absolute and relative concentrations, Fig. 3d) and total organic carbon (TOC) and total nitrogen (TN) content in the sediment separated the samples from spring 2015 from the other seasons. In addition, biomass of macrobenthos excluding the large burrowing bivalve *Laternula elliptica* was higher in the samples of Faro and Creek than at Isla D (Supplementary Fig. 2). A similar gradient is evident when contrasting winter samples and glacier front (Isla D) samples to summer and winter 2015 and spring 2016. In the PCA, these winter and Isla D samples cluster clearly opposite from the samples farther from the glacier (Faro and Creek) (Fig. 4). At the glacier front site Isla D, the sediment was silty (Supplementary Table 3) and contained high biomass of meiofauna and *L. elliptica* (Supplementary Figs. 2 and 3). CR was lowest in these samples (least negative; Fig. 3b). Temporal changes on their own explain 22% of the variability (Fig. 4, Comp. 2): Summer 2015 and spring 2016 were characterized by higher sea surface and bottom water temperature (Fig. 2f), increased glacial melt in the month prior to the in situ benthic metabolism measurements (Fig. 2d), hence also higher SPM concentrations in the water column (Fig. 2h).

Regression models clearly illustrate that net benthic metabolism (NCM) is enhanced by lower SPM concentrations in the water column (proxies for better irradiance conditions and/or lower sedimentation on the shallow water benthic microalgal assemblage) under low glacial run-off (Supplementary Table 4): with lower SPM concentrations in the sea water (clear waters in spring 2015) and a higher relative chl-*a* content of the sediment (the ratio of chl-*a* to the total pigment concentration in the sediment was for 32% explained by higher benthic microalgal biomass; Supplementary Table 4), a higher NCM is measured (53% of the variance in NCM explained, Supplementary Table 4). In this way, we show here that the combined effect of the spatial and temporal variability in glacial melt run-off leads to strong differences in estimated GPP by the benthic microalgae.

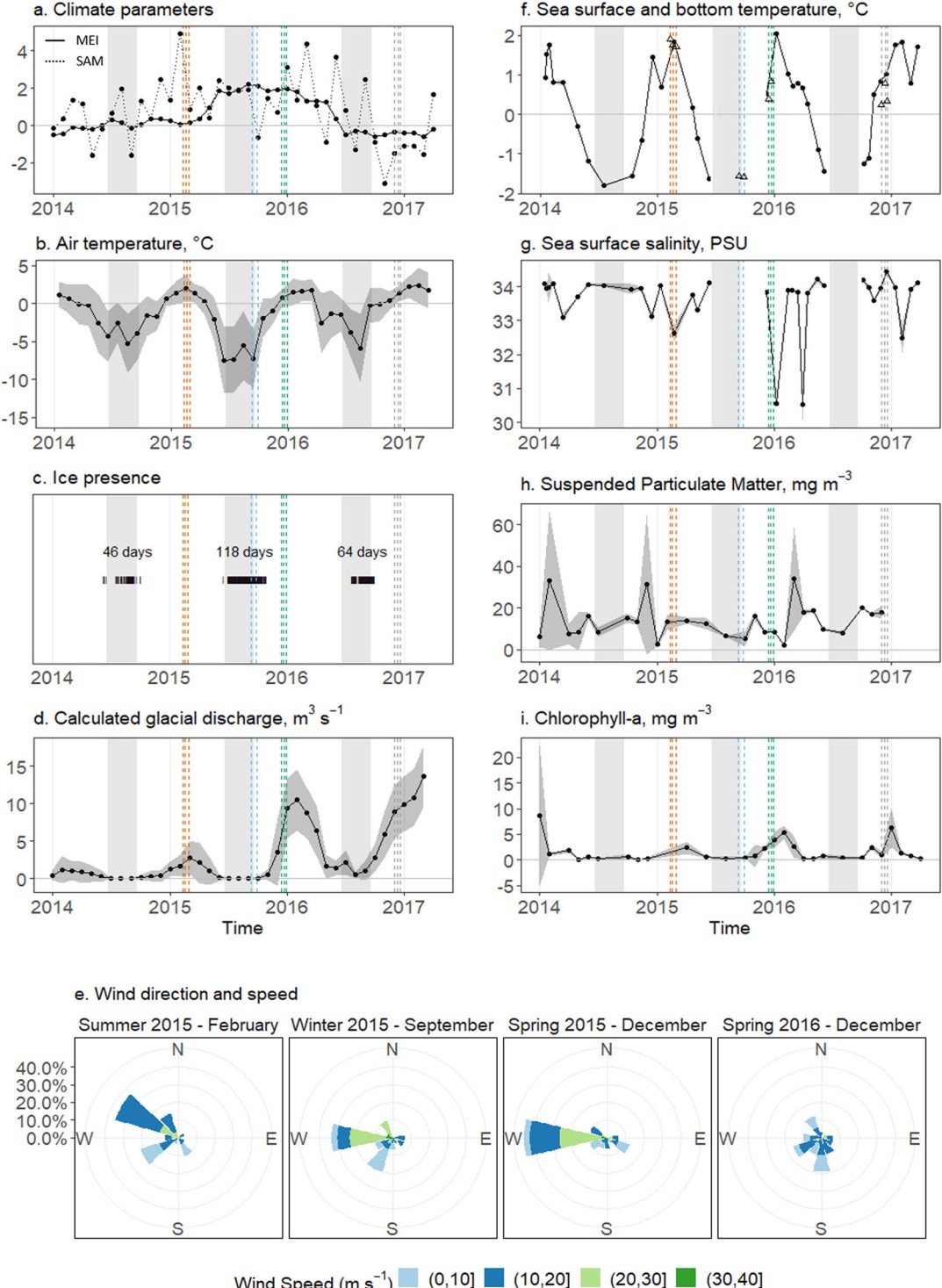

**Fig. 2 Environmental parameters. a** Monthly averages of the multivariate ENSO index (MEI v2) and southern annular mode (SAM), **b** average monthly air temperature, **c** daily sea ice presence, **d** monthly averaged modelled glacial discharge, **e** wind speed and direction during the months of the in situ incubations, with contours indicating the percentage of time the wind had a particular direction, **f** average sea surface temperature (0–1 m water depth) of inner Potter Cove (filled dots),  and average bottom water temperature measured during benthic chamber incubations at each site (open triangles), **g** average salinity, **h** SPM, and **i** chl-*a* in the surface waters (0–1 m water depth) of inner Potter Cove from 2014 to 2017. In panels **a**–**d** and **f**–**i**, grey ribbons represent standard deviation; vertical dashed lines indicate the dates of the in situ incubations in summer 2015 (orange), winter 2015 (blue), spring 2015 (green) and spring 2016 (grey); grey zones indicate the astronomical austral winter; limited data coverage of sea surface temperature and salinity in autumn and winter 2015 and 2016 is shown as disconnected data points. Source data can be found in Supplementary Dataset 1.

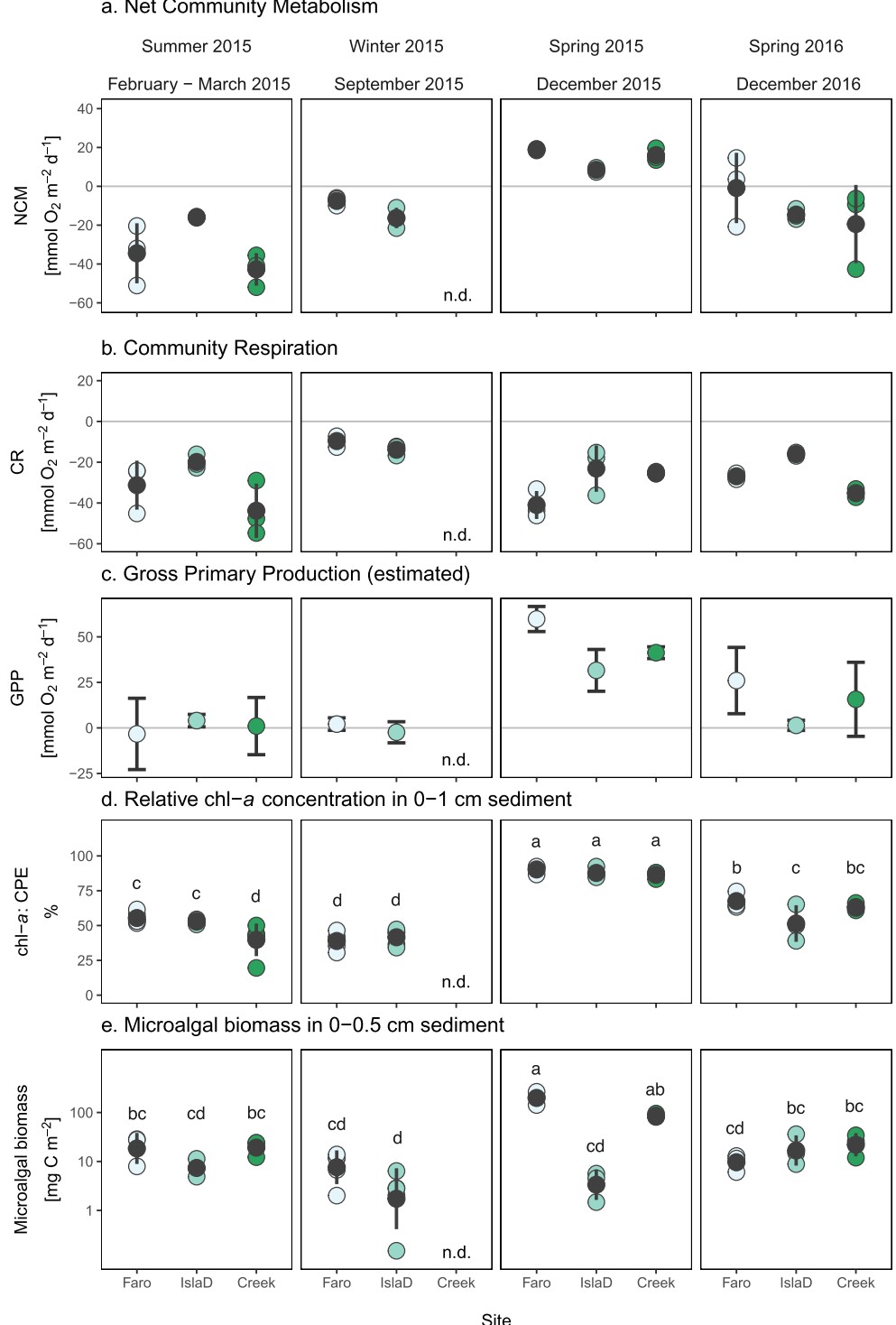

**Fig. 3 Benthic metabolism parameters.** Temporal variability in benthic parameters in the three investigated sites in terms of **a** Net community metabolism (NCM) measured in the transparent chambers ($n = 3$, but see three omissions in the section "Methods"), **b** community respiration (CR) measured in the dark chambers ($n = 3$, but see two omissions in the section "Methods") and **c** calculated gross primary production (GPP) based on NCM and CR (see the section "Methods"), **d** chlorophyll-*a* as a percentage of total chloroplastic pigment equivalents (CPE) in the upper 1 cm sediment ($n = 5$–6 in summer and winter 2015 and $n = 3$ in spring 2015 and 2016) and **e** benthic microalgal biomass in the upper 0.5 cm sediment (on $\log_{10}$ scale; $n = 3$, except for winter 2015 samples, where $n = 5$) in the three investigated sites. Individual measurements are shown as coloured points, while average ± s.d. is shown in black. Letters in **d** and **e** refer to statistical differences that are specified in Supplementary Table 2. Source data can be found in Supplementary Dataset 2.

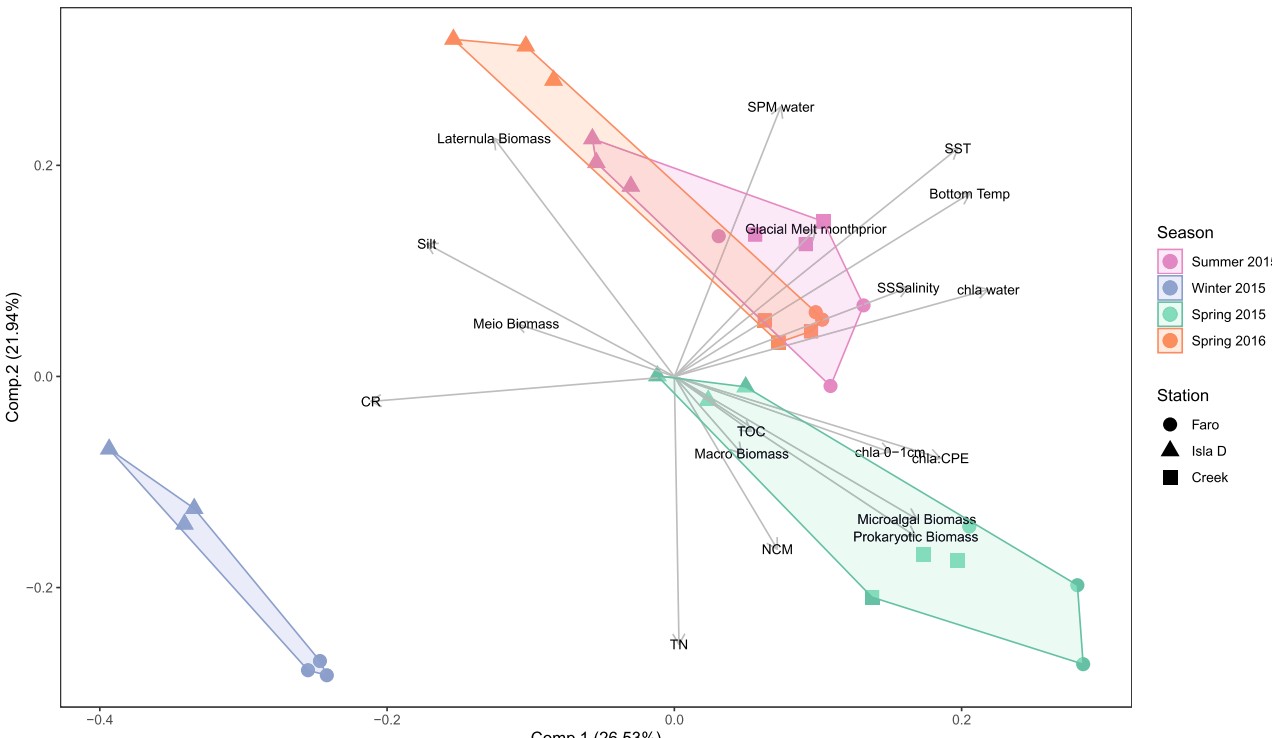

**Fig. 4 PCA of the water column and benthic parameters.** Benthic parameters include environmental, biotic (averaging macrofauna and Laternula elliptica data over summer and spring 2015 because of lacking winter data) and benthic carbon balance parameters. Note the temporal clustering and the separate cloud of samples close to the glacier front (Isla D; triangles). Source data can be found in Supplementary Dataset 3.

Following a 2.5-fold stronger glacial discharge in spring 2016, estimated GPP by the benthic microalgae in the same season was reduced by a factor 2–2.5 in locations farther from the glacier as compared to spring 2015, whereas the sites most impacted by glacial discharge (Isla D) displayed a 22-fold reduction in estimated GPP.

Glacial melt-related disturbance in Potter Cove is likely one of the factors strongly shaping the overall benthic communities. Filter-feeder dominated macrobenthic communities are found in fine sand areas with little glacial disturbance, while scavenger and opportunistic feeder dominated communities, in contrast, are present where frequent ice scouring induces mortality in fauna inhabiting muddier sediments[11]. These patterns also result in overall lower benthic production potential[22] and lower mineralization rates[39] in areas closer to the glacier front. In addition, increased turbidity close to the glacier front is known to affect primary production of phytoplankton[36] and macroalgae[21] communities. On the other hand, macroalgae have shown high colonization potential of the newly ice-free areas[8,9]. Combined with a possible increase in PAR duration in spring due to a decrease in annual fast ice duration[40], a higher overall macroalgal productivity can be expected[21,40].

If benthic net carbon respiration remains higher than primary production on longer time scales, a reduction in the available biomass for benthic consumers might be expected[41,42]. Lower food availability, in addition to increased ice scouring and sediment inputs during high glacial melt discharge periods, can result in a lower secondary production as seen in the reduced macrofauna biomass in spring 2016 (Supplementary Fig. 2). An exception to the joint decline in benthic primary production and macrofauna biomass is the high recruitment of *L. elliptica* (cf. high densities, small individual biomass in Supplementary Fig. 3, Supplementary Dataset 6, Fig. 5) at the glacier front (Isla D) in spring 2016. This large burrowing bivalve is known to graze very

efficiently on (resuspended) benthic diatoms[43]. Hence, the very abundant local *L. elliptica* population might have acted as a top-down control on the relatively low biomass of benthic microalgae close to the glacier front (Isla D) during spring 2016. In addition, in contrast to non-burrowing macrofauna, the juveniles of *L. elliptica* are known to be resistant to sediment deposition[44] resulting from increased glacial melt. The same authors, however, warn about a possible decrease in mean population lifespan of the species under persistent glacial disturbance, with a risk on losing its key role in the benthopelagic carbon drawdown in areas of high sediment deposition.

**Outlook for coastal WAP under increasing glacial melt.** The seasonal series of in situ measurements of benthic metabolism in combination with different climatological conditions is unique for the region: around Antarctica, there are to our knowledge no other studies on the carbon balance in shallow benthic communities with both spatial and temporal resolutions. Our ecosystem view from climatological drivers, and complementary responses in cryosphere and oceanography was instrumental in explaining temporal patterns in benthic primary production. In addition, the strong interannual contrasts between El Niño and La Niña conditions in this dataset offer an outlook to future climate change scenarios in other Antarctic shallow soft sediment systems. Global climate model projections under continued greenhouse gas emission scenarios up to the year 2100 (RCP 8.5) suggest an increased occurrence of +SAM[45]. Combined with La Niña conditions, these conditions favouring increased glacial melt, could represent the near and mid-term future of coastal WAP areas. The influence of glacier-originated and other suspended particles is an extended phenomenon along the WAP waters (i.e., see Fig. 6 in Fuentes et al.[46]). With glaciers further south on the WAP retreating landward[47] and continuous atmospheric warming-

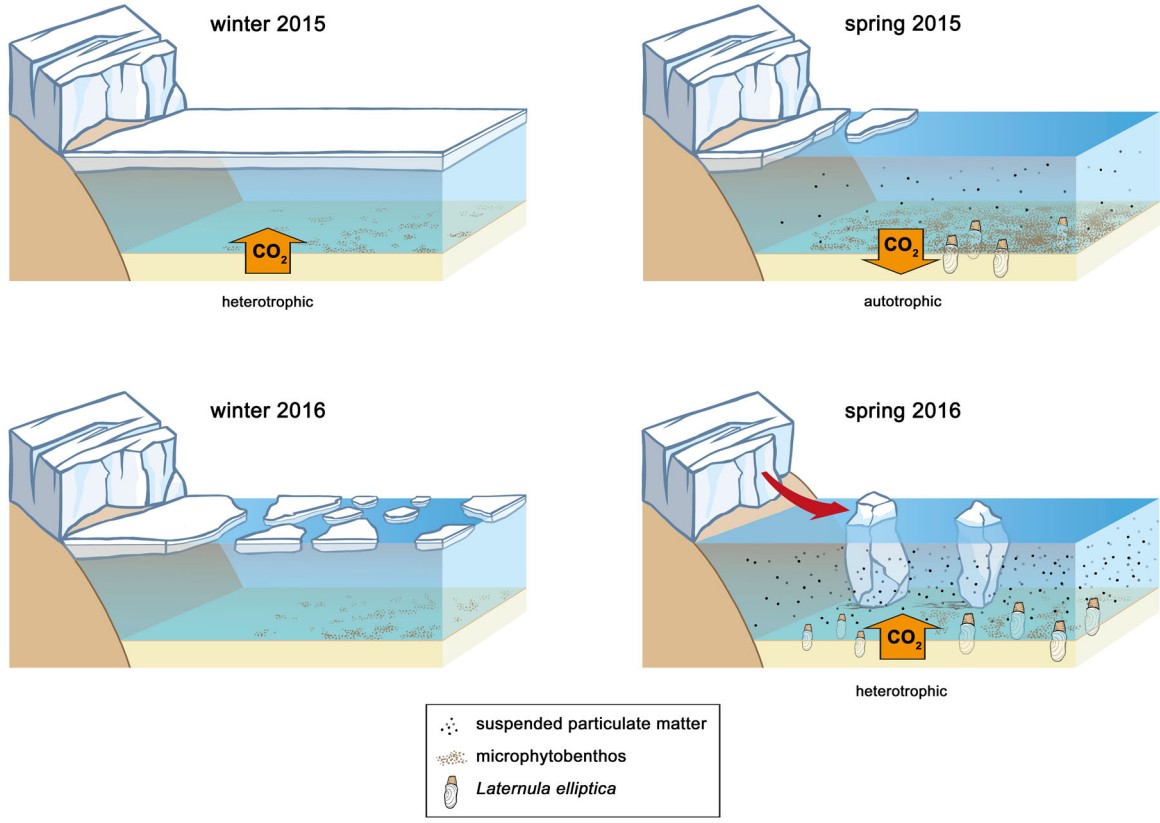

**Fig. 5 Schematic overview of the influence of climatic conditions on the marine ecosystem of Potter Cove.** El Niño conditions in winter and spring 2015 reflect a 'past scenario' and La Niña conditions in winter and spring 2016 reflect a possible 'future scenario'. Benthic metabolism is indicated as difference between gross primary production (GPP) and community respiration (CR).

induced glacial melt, the sediment-laden meltwater plumes are likely to steer these fjords and bays progressively more towards net heterotrophy as observed for Potter Cove (Fig. 5).

In addition, increases in ice-scour-induced mortality to shallow benthic communities are expected as a result of shorter sea ice seasons[48] that allow icebergs to reach coastal areas. These processes would lead to an increased return of $CO_2$ into the atmosphere through decomposition of dead material. Given the evident fragility of these coastal Antarctic ecosystems already under climate change pressure, they should be protected against additional anthropogenic disturbance such as tourism to maintain their important role in the whole system's biogeochemistry and food web.

## Methods

**Study site**. Potter Cove is a ~3 km long and 1.2 km wide, shallow, fjord-like bay in the south-west of King George Island/Isla 25 de Mayo, an island located at the tip of the Antarctic Peninsula (Fig. 1). The cove receives freshwater input from the Fourcade glacier[5] and from seasonal meltwater discharge as a consequence of permafrost and snow melt. Sediment accumulation from turbid meltwater plumes discharging in Potter Cove[24] has almost tripled since the 1940s (<0.15 to 0.15–0.45 g cm$^{-2}$ yr$^{-1}$)[49], during the Fourcade glacier transition from a tidewater to a land-terminated glacier[50]. A dominant clock-wise circulation, with an average current speed of 0.03 m s$^{-1}$[51], transports suspended matter out of the cove. In the shallowest areas of Potter Cove, the water column turbidity is further sustained through wind-induced or waved-induced sediment resuspension[20]. The three locations included in the present study (6–9 m water depth, Fig. 1, Supplementary Table 1, Supplementary Dataset 7) are located in the inner part of the cove and are mainly characterized by soft sediment[11,52]. The locations, namely Creek, Faro and Isla D, are all glacial ice-free[5], but are regularly covered by sea ice during winter[24]. Isla D is the closest to the glacier front and the most recently ice-free area (2003–2006). Faro lies near the northern shore of the cove and became ice-free between 1988 and 1995. Creek is adjacent to a seasonal meltwater river ("Potter Creek") and has been ice-free since the early 1950s. All investigated sites can be classified as meltwater fjord habitats[53] but owing to their location and the current system in Potter Cove, they were permanently and naturally exposed to contrasting intensities of

disturbance, a consequence of the turbidity and the sediment accumulation regime[11,21]. The amount of SPM in the water column is highest at Isla D and Creek and lowest at Faro[50]. As a result, sediment accumulation is highest at Isla D, intermediate at Creek and lowest at Faro[11,54]. Sample processing was conducted at the Argentinian-German Dallmann Laboratory at the Argentinian Carlini scientific research station, located at the southern shore of Potter Cove.

**Climate and meteorological data**. Climate variability indicators used in this study are the Multivariate ENSO Index (MEI) and SAM indices. The monthly averaged MEI indices were retrieved from the National Oceanic and Atmospheric Administration Earth System Research Laboratory website (http://www.esrl.noaa.gov/psd). The monthly SAM index was obtained from the British Antarctic Survey (BAS, http://www.nerc-bas.ac.uk/icd/gjma/sam.html). Air temperature, wind speed and direction were measured every 3 h by the Servicio Meteorológico Nacional (SMN) of the Argentine Air Force at Carlini Station.

The period and the annual number of days with ice cover was calculated for Potter Cove using daily photographic observations of the presence or absence of ice cover between 2014 and 2017[55–57].

**Water column sampling**. The oceanographic sampling is part of a long-term monitoring programme that started in 1991, during which the water column in Potter Cove is sampled weekly in summer and bi-weekly in winter, whenever the meteorological conditions allow it at several sampling stations in the Cove. A Sea-Bird SBE CTD (Sea-Bird Electronics Inc., USA) was used for data on seawater temperature and conductivity (transformed in salinity). Average sea surface temperature (SST) and salinity were extracted from the upper meter from the long-term monitoring station in inner Potter Cove (E1[24]). Bottom water temperature (6–9 m water depth) was recorded with HOBO Pendant loggers (Onset, Bourne, USA) during in situ benthic chamber deployments (see 'Community metabolism measurements'). Water samples were collected at four depths (i.e., 0, 5, 10 and 20 m) at the long-term monitoring station in inner Potter Cove with 4.7 L Niskin bottles. For chlorophyll-$a$ (chl-$a$) analysis, seawater (0.25–2 L) was filtered through 25 mm Whatman GF/F filters under gentle vacuum and dim light. Chl-$a$ was extracted in 90% acetone for 24 h at 4 °C in the dark. Absorbance was measured using a Shimadzu RF-1501 spectrophotometer, and concentrations were calculated and corrected for phaeopigment content[58]. Total SPM was measured gravimetrically after filtering 0.25–2 L seawater through combusted pre-weighed 25 mm Whatman GF/F filters. After filtration, filters were rinsed twice with distilled water

to remove salts, then dried for 24 h at 60 °C, and weighed again. Chl-*a* and SPM data were integrated over 0–20 m.

**Glacial melt model**. Falk et al. [30] adapted the glacial melt model[59], based on solving the surface energy balance equation on a spatially distributed grid of the hydrological catchment area draining into the Potter creeks and Potter Cove. The Potter Creek catchments have altogether an area of 8.423 km² and are glaciated to 82%. Meteorological data from an automated weather station installed between 2010 and 2017 on the Fourcade glacier was used to drive the glacial melt model, as well as the glaciological surface mass balance measurements during that period. An important and difficult task is to define the capture zone of subglacial and glacial run-off. The Potter Creek river basins were defined based on the topography and drainage networks survey[60]. It is difficult to generalize observations of magnitude and frequency of hydrological flow events because there is a lack of long-term data. A hydrological field study was carried out in austral summer 2010/2011. The monthly discharge run-off in the Potter Creeks were calculated to $Q_{obs} = 1.13$ hm³ month⁻¹ for South Potter creek and $Q_{obs} = 1.43$ hm³ month⁻¹ for North Potter creek. The groundwater flow was estimated to $Q_{obs} = 1.21$ hm³ month⁻¹ for the summer months January and February. A simple calculation allows to determine that a total mixed discharge (run-off and groundwater discharge) is $Q_{obs} = 3.77$ hm³ month⁻¹, of which 70% corresponds to the discharge run-off and 30% to groundwater discharge[60]. The glaciological model simulations carried out by Falk et al. [30] are well in agreement with the total meltwater discharge estimated by hydrological observations and a simplified hydrological model[30].

**Benthic sampling**. We measured biogeochemical fluxes across the sediment–water interface and sampled the sediment to characterize benthic communities and environmental parameters during four field campaigns in summer 2015 (February–March), winter 2015 (September), spring 2015 (December) and spring 2016 (December) at three sites with contrasting glacial melt influence (Fig. 1) at 5–9 m water depth (Supplementary Table 1). Although strictly not in winter anymore, for the sake of simplicity, we considered the sampling at Faro on 28 September 2015 as a winter sampling since it was performed only 2 weeks later than the winter sampling at Isla D and the cove was still densely covered with sea ice. Similarly, we still consider the sampling at Faro on 28 December 2015 as a spring sampling, since it was performed in the same expedition as the spring sampling at the other two sites. In winter 2015, the cove was fully covered with ~1 m-thick sea ice and, since opening a hole in the ice sheet manually was very labour intensive, the number of sampled sites was limited to two (Faro and Isla D).

**Sediment properties**. For the determination of sediment properties and biogenic sediment compounds, sediment was sampled with 3.6 cm diameter plexiglass cores in three to five replicates by SCUBA divers. Sediment subsamples were taken with cut-off syringes (cross-sectional area = 1.65 cm²) and sliced in 1 cm intervals down to 5 cm sediment depth. Each interval was analysed for various parameters including median grain size, porosity, photosynthetic pigments, TOC and TN. Sediment samples for photosynthetic pigments were stored in the dark at –80 °C within one hour after processing. Samples for the analysis of other sediment parameters were stored at –20 °C.

The median grain size was determined with a Malvern Mastersizer 2000G, hydro version 5.40, which uses a laser diffraction method and has a measuring range of 0.02–2000 μm. Sediment porosity was estimated after drying sediment samples over a period of at least 2 days at 105 °C and further calculated according to Burdige (2006)[61]. The TOC and TN contents were measured by combustion using an ELTRA CS2000 with infrared cells after acidifying the sample (3 mL of 10 M HCl).

Chl-*a*, phaeophytin plus phaeophorbid (phaeo) pigment concentrations were determined in the home lab in Belgium by HPLC (Gilson)[62] after all expeditions except for winter 2015. In winter 2015, a –80 °C freezer was not available, so the samples were extracted immediately on site and measured spectrophotometrically according to Lorenzen (1967)[63]. When measured using chlorophyll pigment standards, the two methods are comparable in terms of pigment concentrations, but the Lorenzen method generally results in 6–9% higher chl concentrations than the HPLC method[64]. The comparison of the two methods on pigments from sediment samples is likely more complicated. We show chl-*a* concentrations without correction factor, but acknowledge that the chl-*a* concentrations from winter 2015 might be ~10% overestimated. The bulk of pigments (chl-*a* plus phaeo) was termed chloroplastic pigment equivalents (CPE). The ratio of chl-*a* to CPE (% chl-*a*) serves as a quality indicator of the labile organic matter in the sediment.

### Benthic community structure

*Prokaryotes*. For prokaryotic density determination, the same sampling and sub-sampling approach was used as for the sediment properties. Each 1 cm interval of the sediment core was fixed in a 2% formaldehyde/seawater filtered solution and stored at 4 °C. Prokaryotes in the subsamples were stained using the acridine-orange-direct-count method[65] and subsequently counted with a microscope (Axioskop 50, Zeiss) under UV-light (CQ-HXP-120, LEj, Germany). The method does not differentiate between groups of bacteria or archaea. For each subsample,

single cells were counted on two replicate filters and for 30 random grids per filter (dilution factor 4000). Prokaryotic biomass was estimated by the determination of the mean prokaryotic cell volume in the first two centimetres of the sediment with a "New Portion" grid (Graticules Ltd, Tonbridge, UK)[66], converted into biomass using a conversion factor of $3.0 \times 10^{-13}$ g C pm⁻³,[67] and multiplied with the pro-karyotic density. Thereby, each mean prokaryotic cell volume represents the mean of 100 counted grids.

*Benthic microalgae*. For benthic diatom biomass, the upper 0.5 cm sediment layer of a 3.6 cm diameter core was transferred into a scintillation vial and 5 mL GF/F filtered seawater (Whatman, UK) and 1 mL of 25% glutaraldehyde were added. The vial was stored at 4 °C until further analyses. Diatoms, the major components of benthic microalgae in the study area[68–70], were identified and counted. In valvar view, the length and width of pennate valves and the diameter of centric valves were measured. With an assumed height of 1 μm[71], diatom cell biovolume was calculated[72,73]. The diatom cell biovolumes were converted into diatom carbon biomass using the following conversion formula: pg C cell⁻¹ = 0.288 × cell volume$^{0.811}$ [74]. The samples from December 2016 have been used for a more thorough identification in the work of Hoffmann et al. [22], in which a slightly different protocol was used: diatom valves were cleaned with 30% hydrogen per-oxide and, after proper rinsing with deionized water, mounted in Naphrax after Al-Handal and Wulff[68]. Enumeration of diatom valves on the slides was made by counting intact valves on the whole slide using a Zeiss Axio Image 2 compound microscope equipped with differential interphase contrast under 400-fold magni-fication. Although diatoms can be identified to the species level, very large cells, e.g. of *Gyrosigma* sp. can be overlooked with the applied method (e.g. due to low abundance, lost during sample dilution), hence not counted. Therefore, diatom density and hence diatom biomass data of December 2016 are likely somewhat underestimated (see discussion in Hoffmann et al. [22]).

*Meiofauna*. For the determination of meiofauna density and biomass, between three and five sediment samples were also taken with 3.6 cm diameter Plexiglass cores. Sediment samples of the first five centimeters were stored in a 4% for-maldehyde/seawater filtered solution at 4 °C until extraction at the home laboratory in Belgium. The samples were sieved over a 1 mm and 32 μm mesh, then cen-trifuged three times in a colloidal silica solution (Ludox TM-50) with a density of 1.18 g cm⁻³ and stained with Rose Bengal[75]. Afterwards, benthic meiofauna was identified on higher taxon level (order or class) and counted. To determine the meiofauna biomass, the TOC content of single taxa was measured with a FLASH 2000 NC Elemental Analyzer (Thermo Fischer Scientific, Waltham, USA). Calci-fying organisms were acidified prior to the analysis.

*Macrofauna*. The benthic macrofauna was sampled with a Van Veen grab (530 cm² surface area). At each location, three to four sediment samples were sieved over a 1 mm mesh and stored in seawater-buffered 4% formaldehyde/seawater solution. In the laboratory, the taxa were identified to the lowest possible taxonomic level (at least family level), counted and weighed (bivalves without shell, except for the very small and brittle specimens). Biomass was estimated as ash-free dry weight (AFDW), which was determined by subtracting the ash weight (after combustion at 500 °C) from the dry weight (dried for 48 h at 60 °C). AFDW was converted into carbon by assuming that 50% of the AFDW is carbon[76]. The Van Veen grab sampling results in a strong underestimation of the density of the large Antarctic bivalve *L. elliptica*. Therefore, two transects of eight grids (45 cm × 45 cm) were randomly placed on the seafloor by SCUBA divers and photos were taken (Nikon D750 with a rectilinear Nikon 16–35 mm lens in a Nauticam underwater housing and two Inon Z-240 strobes). Unclear photos were omitted. At all sites and occasions, between 10 and 16 photos were used to count siphons of *L. elliptica* to determine the density and to measure the siphon width (maximum distance between outer edges of the two siphons of one individual, calibrated against the frame size). Assuming a linear relationship between siphon width and AFDW, biomass of *L. elliptica* was estimated from the siphon width–AFDW conversion factor calculated in Hoffmann et al. [39].

**Community metabolism measurements**. The methods to quantify in situ benthic community metabolism have been broadly described in Hoffmann et al. [39]. Briefly, SCUBA divers carefully deployed three transparent and three black chambers (inner diameter 19 cm, height 33 cm) into the sediment at each location. Approx. 15 cm of sediment and 18 cm of overlying water were enclosed. Cross-shaped stirrers powered by a 12 V lead-acid battery kept the overlying water homogenous during the incubation, gently enough to avoid sediment resuspension. The incu-bations lasted 20–22 h and included light and dark periods. Owing to dive security regulations (e.g. danger for wandering ice growlers and leopard seals), we could not perform measurements after sunset to distinguish light from dark exchange rates. Therefore, the resulting exchange rates are daily net rates.

The enclosed overlying water in the chambers was sampled at the start and end of each chamber incubation, using gas-tight glass syringes. Water samples were kept in darkness and at ambient cold air temperature until further processing (within 1 h after sampling). Winkler titration was used to determine the oxygen concentration in the water sample in technical duplicates on site[77]. The TOE rate by the benthic community during the incubation was calculated by standardizing

the change of oxygen concentration over time for the enclosed overlying water volume and the area of exchange at the sediment surface. The volume of the overlying water was calculated by using the average height between the seafloor and the chamber lid, measured after starting the incubations at five locations of each chamber. TOE measured in the dark chambers (negative) is further referred to as CR, whereas the TOE measured in transparent chambers is equivalent to the NCM (net result of the oxygen produced and consumed during daylight hours and oxygen that is consumed during dark respiration). NCM is positive when more oxygen is produced than consumed and negative when more oxygen is consumed than produced. GPP is then calculated as the sum of average NCM (of triplicate transparent chambers) and the absolute value of average CR (of triplicate dark chambers) per site and sampling occasion. GPP can be negative due to the way it is calculated when respiration exceeds production, such as when organisms consume previously accumulated stock. Similarly, close to zero values reflect the balance between both processes. Nevertheless, as GPP is photosynthetic rate, any values below zero were just assumed to be zero. Three values of NCM and two values of CR were omitted from further analysis because of technical issues during incubation (one transparent chamber at Isla D in summer 2015, one transparent chamber at Isla D and one at Faro in spring 2015, one dark chamber at Faro and one at Creek in spring 2016).

**Statistics and reproducibility**. To test for spatio-temporal variability in the selected parameters, a two-way ANOVA with fixed factors season and station was applied, followed by Tukey HSD post-hoc tests in case of significant effects. If assumptions for parametric tests were not met (normal distribution, homogeneity of variances), semi-parametric Permanova with subsequent pairwise comparisons were applied. Sample sizes differ depending on type of measurement (min. $n = 3$, but $n = 4–6$ where possible) and are specified in Fig. 3, Supplementary Fig. 3 and Supplementary Tables 2–4. Patterns in water column and benthic parameters were explored with principal components analysis (PCA). Water column parameters were included in the PCA. Water column data from the dates closest to the benthic sampling date were selected. Since these data are derived from a single station within inner Potter Cove, the same value was used for comparison with all benthic sites. Since macrobenthos and *L. elliptica* could not be sampled in winter 2015, and PCA cannot cope with missing values, we used averaged macrobenthic and *L. elliptica* parameters (over summer and spring 2015) to represent winter 2015 in the PCA. Hence, we assumed that the macrobenthic and *L. elliptica* populations remained stable during winter. Both Arctic[78,79] and Antarctic studies[80] have shown that this is a reasonable assumption to make. Linear models were constructed to regress NCM on abiotic and biological parameters. To identify the best predicting parameters, a preselection of predictor parameters was performed using the R package "glmnet"[81]. The remaining predictor parameters were used in a linear model, checked for multicollinearity, and stepwise excluded if they exceeded a variance inflation factor-value of 10. Through backward selection, models with only significant partial regression coefficients were obtained ($p$-value < 0.05). Assumptions were verified visually, and normality of residuals was confirmed with a Shapiro test. All statistical tests were performed using R Statistical Software (version 3.5.2, R Core Team, 2018) using the packages vegan[82], car[83] and glmnet[81]. Results are expressed as means ± s.d. unless stated differently (Supplementary Table 2).

**Reporting summary**. Further information on research design is available in the Nature Research Reporting Summary linked to this article.

## Data availability

Data are publicly available at PANGAEA Data Publisher for Earth & Environmental Science (www.pangaea.de), doi: 10.1594/PANGAEA.926615.

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

## Acknowledgements

We are grateful to the Instituto Antártico Argentino (IAA) and the crew at Carlini Station for the logistic support during the field campaigns. We also warmly thank Oscar González, Doris Abele and the Argentinian Army (Ejército Argentino) divers of CAV 2014–2015, the overwintering divers (José Luis Müller, Ezequiel Tulián, Ramón Alfredo Torres, Fernando Cumil) and scientist Francisco Ferrer of CAI 2015, the crew of CAV 2015–2016 and 2016–2017 and scientific divers Elisa Merz and Christopher Brunner, who

made this study possible. We would like to thank Volker Asendorf and Volker Meier for their help with benthic chamber preparation, Bart Beuselinck and Dirk Van Gansbeke for laboratory analysis of sediment and water properties, Simon Schnebert and Annick Van Kenhove for prokaryotic counts, Guy De Smet for meiofauna counts and Bob De Clercq from UGent FIRE statistical consulting. The schematic summary in Fig. 5 is the work of Dr. Hendrik Gheerardyn (www.hendrikgheerardyn.com). The first author is a senior post-doctoral research fellow at Research Foundation—Flanders (FWO Belgium) (grant nos. 1201716N and 1201720N). The present manuscript presents an outcome of the EU research network IMCONet funded by the Marie Curie Action International Research Staff Exchange Scheme (FP7 IRSES, Action No. 318718), the vERSO project (www.versoproject.be) funded by the Belgian Science Policy Office (BELSPO, contract no. BR/132/A1/vERSO), the SenseOCEAN project (www.senseocean.org) funded by the European Union (FP7, grant no. 614141) and the PACES programme funded by Alfred-Wegener-Institut Helmholtz-Zentrum für Polar- und Meeresforschung.

## Author contributions

U.B. designed the study; U.B., F.P., R.H., A.T. performed the measurements; U.B., F.P., R.H., S.V., N.L., A.W. and A.A. collected and analysed data; I.R.S. provided the ocea-nographic data, U.F. modelled glacial melt run-off, D.D. provided the sea ice presence data; F.W. and A.V. provided necessary equipment and technical expertise; U.B. wrote the manuscript with substantial input from all co-authors.

## Competing interests

The authors declare no competing interests.
