## [Transparent Peer Review File · Communications Biology]

Reviewers' comments:

Reviewer #1 (Remarks to the Author):

Braeckman and coauthors conducted a study to investigate the potential carbon storage and sequestration capabilities of shallow seafloor habitats in the West Antarctic Peninsula (WAP). The authors present a novel dataset on seafloor metabolism measurements performed during four measurement campaigns covering different seasons in 2015 and 2016. Additionally, the authors spend considerable effort documenting complementary benthic biotic parameters such as biomass of microbes and fauna. The authors identify differences in biotic and metabolic measurements which they ascribe to changing water column turbidity. The authors provide evidence, in turn, that the changes in turbidity arise from melt of nearby glaciers, and that the amount of melt varies in time and is related to large-scale atmospheric patterns (El Niño and La Niña), which alter conditions from cold to warm, respectively.

Major comments

Overall, this is a well-written paper with interesting and novel data. The measurements and data processing seem to have been done meticulously, and I can appreciate the effort of working under challenging Antarctic conditions. There is no data on the amount of sunlight reaching the seafloor, despite this being important to the conclusions being drawn here. There is also no data on carbon burial rates in sediments using e.g. ^{210}Pb and sediment C stocks, which I would consider to be important for any carbon burial assessment. My main comments, however, concern how the story is framed, i.e. within the context of carbon storage and sequestration. The authors present this as "...a potentially large biological negative feedback on global anthropogenic CO₂ emissions" (L45-46). As far as I can see, however, the paper lacks key information on the magnitude of the feedback: how large are the affected areas? How great is the potential carbon drawdown? Having delved into the literature, I get the impression that the potential role of the shallow Antarctic benthos for carbon storage and sequestration is very small indeed and while it may have important ecological implications locally, it is rather inconsequential for global carbon cycling. There are other feedbacks that may affect carbon storage negatively which are not discussed. My reasoning is outlined below.

The papers by Barnes are key references. I delved into some of these to assess whether Antarctic blue carbon is indeed a potentially large biological negative feedback on global anthropogenic CO₂ emissions, as stated in the Introduction of this paper and elsewhere throughout the manuscript. First, the width of the continental shelf in the Antarctic is very narrow; the 0-50 m depth band is just 18,400 km² (Barnes 2017). For comparison, the Arctic coastal region is ~6 million km² (Gattuso et al. 2006). The area is therefore very small. Barnes et al. (2020) estimate the carbon "sequestration" capacity of WAP fjords through generation of new zoobenthic biomass to be "> 780 tonnes C yr⁻¹" (the term "sequestration" within the Barnes papers and within the present manuscript is used very loosely; the IPCC defines the term differently, which is confusing). In any case, the ~800 t C yr⁻¹ of new zoobenthic biomass is basically insignificant compared to global CO₂ emissions which are currently at 37 Gt C yr⁻¹. In comparison, global forests draw down 1.1 Gt C yr⁻¹ and macrophytes sequester up to 0.9 Gt C yr⁻¹ (Duarte 2017). In my view, for a carbon sink to be significant, it needs to act on the Gt scale.

There are two potential negative feedbacks to Antarctic Blue Carbon that are not really discussed. The first is benthic disturbance through iceberg grounding and scouring. It is persistent in these areas; every year it impacts a third of the seafloor at depths < 25 m, and for shallower waters it is > 90 % (Barnes 2017). This process will affect the proliferation of zoobenthic communities in newly exposed shallow areas and would disturb buried matter in the sediments. This point is mentioned briefly in the

Introduction and Outlook sections (L53-57; L237-239) but its impact on Blue Carbon is not estimated. The second negative feedback is related to the erosion of marine-terminating glaciers that is required in order to expose new seabed. There is emerging evidence in the Arctic that marine-terminating glaciers support high productivity through rising subsurface meltwater plumes that transport large volumes of nutrient-rich deep waters to the surface, stimulating phytoplankton production (Meire et al. 2017). Fjords with only land-terminating glaciers lack this upwelling mechanism and have lower productivity. Since the carbon burial efficiency depends primarily on the supply of organic matter to the seafloor, it is not clear to what extent this negative feedback would offset Blue Carbon gains.

I struggled with the concept of autotrophic and heterotrophic benthic ecosystems being equated to carbon sinks and sources, respectively (L31-35 and elsewhere). I understand that there can be a net drawdown of CO₂ in autotrophic communities and vice-versa, but the fate of that carbon is ultimately what matters. Since this concept forms the foundations of the paper, there needs to be much clearer evidence that this is indeed the case.

In summary, I believe that this is a novel dataset that has potentially important ecological implications for our understanding of high-latitude systems. However, I believe the 'benthic carbon storage and sequestration' spin on this data should be largely eliminated as it raises numerous fundamental questions and works against what is otherwise a novel and interesting dataset.

Detailed comments

L32: "Carbon sink" usually refers to carbon that is sequestered on geological timescales. Here and throughout the manuscript it would be good to align these terms (e.g. storage, sequestration, sink, burial, etc.) with the 'official' definitions by the IPCC, for clarity.

L45-46: It would be good to include quantitative estimates- how big is the feedback, potentially? Is this locally/globally significant?

L46: are in retreat

L55: and consequently a

L51-56: Here you describe a potential negative feedback, but you do not present it quantitatively within the context of Blue Carbon gains. Which is expected to be more important?

L51-56: Marine-terminating glaciers support high productivity; their reduction could be another negative feedback. See work by Meire et al. 2017 in *Global Change Biology*.

L64: There are multiple covariables that can affect the autotroph-heterotroph balance. Increased turbidity would decrease sunlight transmittance and increase sedimentation, both of which would shift the system towards net heterotrophy.

L99: such as that seen

L100: "...and then only before 2003." Please rephrase.

L143-144: "Apart from some patches at the least glacial-disturbed site Faro..." How was this assessed?

L146-153: Negative GPP doesn't make intuitive sense- I guess this is a sensitivity limitation of the method. There are also many values close to zero- are they significantly different from zero?

L140-153: Seabed light (PAR) data would be important here. Were any attempts made to constrain PAR? Did measurements fail?

L165: "...cluster clearly opposite..." Please rephrase.

L175: "...better irradiance conditions..." You do not have data on this, so please state specifically how you can infer better conditions.

L176: "With" should have a small letter "w"

L178: "...higher relative chl-a content..." Would you expect grazing to play a role here? In areas where grazing is intensive there can be a low standing stock but a high productivity due to high turnover of microalgal biomass.

L194: "...are also translated..." Please rephrase. Perhaps "...also result in...?"

L329-333: "Although meteorologically not correct..." Please rephrase and provide arguments as to why you select this approach.

L353: "...after all expeditions except for winter 2015." You can simply say that the samples were either extracted immediately after collection or stored for later laboratory analysis. If you think that one approach biased the measurements, then this should be stated, but otherwise it is not required.

L394: "overseen". Do you mean omitted?

L467: "...remained stable during winter..." Is this a reasonable assumption to make?

L470: "glmnet" What is this? Is it an R software package?

References

Barnes, D.K.A (2017) Iceberg killing fields limit huge potential for benthic blue carbon in Antarctic shallows. *Global Change Biology*

Barnes, D.K.A. et al. (2020). Blue carbon gains from glacial retreat along Antarctic fjords: What should we expect? *Global Change Biology*

Duarte, C.M. (2017). Reviews and syntheses: Hidden forests, the role of vegetated coastal habitats in the ocean carbon budget. *Biogeosciences*

Gattuso, J.P. et al. (2006). Light availability in the coastal ocean: impact on the distribution of benthic photosynthetic organisms and their contribution to primary production. *Biogeosciences*

Meire, L. et al. (2017). Marine-terminating glaciers sustain high productivity in Greenland fjords. *Global Change Biology*

Reviewer #2 (Remarks to the Author):

Brief summary

In their manuscript the authors combine meteorological, glaciological and hydrological data with measurements of benthic metabolism in three stations of an Antarctic coastal bay, Potter Cove. They succeed in demonstrating that the overall climate change, portrayed in MEI and SAM indices, leading to periods of increased temperature, subsequent glacial melting and more runoff/water turbidity in the bay, may result in an overall shift from previously net phototrophic to future net heterotrophic metabolism in this system. The data presented for 2015/16 indeed are supportive of this change and it does fit the causal cascade of relations demonstrated with long term data, models and climatology. Since carbon sequestration by burial and transfer into the food chain is reduced, at least temporarily, under heterotrophic conditions, the well-founded conclusion is that this specific change in Potter Cove may portray the future development of several shallow coastal Antarctic marine environments.

Overall impression of the work.

My overall impression is very positive and I enjoyed reading the manuscript. The text is written in a very compact way and readily understandable. (Drawing from my own limited experience with publishing in nature, the style is appropriate)

The specific data sets from 2015/16 from Potter Cove are a solid ground of data despite the missing macrozoobenthos data in winter 2015. The assumption used for macrozoobenthos, a stable population, is ok in order to perform the statistical analyses.

The strength of this manuscript comes from the combination with the ENSO Index and Southern Annular Mode indices, the observed melting and glacial retreat, turbidity and long monitoring data series at Potter Cove. Thus overall data represent different, but sufficiently long periods of observations in each case. Hydrographic data at the locations and finally benthic fauna and activity

parameters relating to the three sites, Faro, Isla D and Creek, are merged well into a suited data set, that really only reaches its summed value with the research done at those three locations. Despite the difficulties associated with it, glacial melt model, runoff in both creeks and groundwater flow seem to be captured well and sufficiently robust for the subsequent discussion on consequences for Potter Cove benthic systems.

The authors also dealt well with methodological differences (Chl-a spectrophotometric versus HPLC determination). Statistical testing is straight forward but correct, and augmented by a PCA analyses. I am convinced that there is no (maybe non-published ...) data set that would allow to exercise an equivalent cascade of cause-effect relations in order to argue that the shift towards reduced sequestration and thus the loss of a process abating CO₂ emissions is a likely scenario.

Specific comments

Line 162: What does the word "these" relate to? Prior a gradient in sediment parameters is discussed, but "these" could either relate to far or distant from the glacier. Better write: was higher in Creek samples.

Line 166: Spell Tab instead of Table

Line 167: shouldn't this be Fig. S1 only?

Line 201/202 AND 205/207: This first sentence seems much too general to be true. If net respiration surmounts primary production, there may still be as much food for grazers or predators. Heterotrophy by bacteria may support the CO₂-efflux, increased activity by otherwise constant macro-biomass may still persist. So at least on a short time scale this statement cannot be kept at such a general level. It should be specified.

Also the statement is repeated in 205/207. This is repetitive and should be avoided.

Line 240: "into the atmosphere through respiration". Yes there would be an increased return to the atmosphere. But rather decomposition of dead material than respiration.

Line 313: The sentence starting "An important and ..." does not seem to continue correctly, particularly the "and" somehow does not seem to fit the message.

Line 425-128: I am missing the explicit information as to which individuals of *Laternula* were weighed for the siphon width-AFDEW relation either missing or not expressed sufficiently obvious.

I consider all tables and figures necessary and of high quality and appropriate contents, including supplementary material.

Reviewer #3 (Remarks to the Author):

This manuscript is very interesting. It combines climatology, glaciology, oceanography, and benthic biogeochemistry to construct a narrative about how an ecosystem responds to different climatic forces (an El Nino cold winter vs a La Nina warm winter) which offers insight to how climate change could impact the ecosystem's function in terms of carbon storage. The manuscript is comprehensive but not over-long. I think it is worthy of publication in *Communications Biology*. I have listed a few comments below that would help the manuscript before potential publication.

My biggest concern with the entire manuscript is the term "blue carbon". It is immediately used in the text, but never circled back to until the very last sentence of the Outlook section. Blue carbon is such a buzz word, and I would suggest that the manuscript is just as strong without invoking this term and simply referring to "carbon sources" and "carbon sinks", as is done in the title. It is usually used in the context of salt marsh, mangroves, and seagrasses, but I see by the reference titles that the Antarctic

coastal ecosystems have been referred to as possessing blue carbon despite no vegetative ecosystem that can bury recalcitrant organic matter. Can you elaborate on the usage of this term for the WAP? It appears some of the references (particularly the Barnes publications) have previously used this term. Is there ample sedimentation that occurs to bury C before remineralization? Can you provide a range of sedimentation/burial rates that occur for the area (cm yr⁻¹) and carbon sequestration rates (g C m⁻² yr⁻¹)? Can you comment on the mechanisms that preserve algal C against decomposition processes so that burial of C does occur? I would argue that the "blue carbon" term is used so that ecosystem services of threatened coastal habitats could be monetized so their protection is incentivized, but perhaps this is a myopic view of the term. It is not to say that any marine ecosystem cannot sequester/bury/store carbon for durations that could potentially interrupt the carbon-climate change feedback loops; indeed studying how an ecosystem is a sink or source of carbon is extremely relevant. Again, I suggest the authors take out this word from the few instances where it occurs. But if they feel strongly about including this study with the other blue carbon literature, then perhaps they could justify its use by briefly addressing some of the above questions for readers like me.

L43-44: too many commas. "...host productive benthic communities, which in combination with a relatively high burial rate in fjord systems contributes..."

Fig 2. Make axis text bigger please. It would be helpful to have season labels, perhaps at the top of the graph columns, that denote where "summer 2015", "winter 2015", "spring 2015", and "spring 2016" occur. It is a little confusing when looking between text to figure since spring occurs almost on top of the New Year.

L174-181: Since the study doesn't measure irradiance/PAR directly, it is not exactly correct to invoke it here. Could you instead say something like "the measured parameters that are proxies for PAR/irradiance attenuation"?

Table S4: "SSSalinity" should be defined somewhere. Unless it is a typo?

L208: "at at"

L106: add a qualifier like "until later that summer..." regarding the glacial water discharge since a few months later it was quite high.

L018-111: The data support this claim; however, Dec 2016-Jan 2017 is also had equally high chlorophyll concentrations. Can you comment on this occurrence since the 2016 winter was so different but still resulted in high water column chlorophyll?

L129-133: The austral summer of Dec 2016-Jan 2017 appears to have the same magnitude discharge as the austral summer of Dec 2015-Jan 2016. The text explanation confused me claiming that the discharge in Dec 2016-Jan 2017 was 3x higher than any preceding summer. But clearly the summer of Dec 2016-Jan 2017 was approximately equal. Please clarify the text here.

Dear editor and reviewers,

Below, we answer to each of the reviewers' comments in a point-by-point fashion. Newly added text to the manuscript is *italicized* and the line numbers refer to the lines in the revised manuscript.

Reviewers' comments:

Reviewer #1 (Remarks to the Author):

Braeckman and coauthors conducted a study to investigate the potential carbon storage and sequestration capabilities of shallow seafloor habitats in the West Antarctic Peninsula (WAP). The authors present a novel dataset on seafloor metabolism measurements performed during four measurement campaigns covering different seasons in 2015 and 2016. Additionally, the authors spend considerable effort documenting complementary benthic biotic parameters such as biomass of microbes and fauna. The authors identify differences in biotic and metabolic measurements which they ascribe to changing water column turbidity. The authors provide evidence, in turn, that the changes in turbidity arise from melt of nearby glaciers, and that the amount of melt varies in time and is related to large-scale atmospheric patterns (El Niño and La Niña), which alter conditions from cold to warm, respectively.

Major comments

Overall, this is a well-written paper with interesting and novel data. The measurements and data processing seem to have been done meticulously, and I can appreciate the effort of working under challenging Antarctic conditions. There is no data on the amount of sunlight reaching the seafloor, despite this being important to the conclusions being drawn here. There is also no data on carbon burial rates in sediments using e.g. ^{210}Pb and sediment C stocks, which I would consider to be important for any carbon burial assessment. My main comments, however, concern how the story is framed, i.e. within the context of carbon storage and sequestration. The authors present this as "...a potentially large biological negative feedback on global anthropogenic CO₂ emissions" (L45-46). As far as I can see, however, the paper lacks key information on the magnitude of the feedback: how large are the affected areas? How great is the potential carbon drawdown? Having delved into the literature, I get the impression that the potential role of the shallow Antarctic benthos for carbon storage and sequestration is very small indeed and while it may have important ecological implications locally, it is rather inconsequential for global carbon cycling. There are other feedbacks that may affect carbon storage negatively which are not discussed. My reasoning is outlined below.

The papers by Barnes are key references. I delved into some of these to assess whether Antarctic blue carbon is indeed a potentially large biological negative feedback on global anthropogenic CO₂ emissions, as stated in the Introduction of this paper and elsewhere throughout the manuscript. First, the width of the continental shelf in the Antarctic is very narrow; the 0-50 m depth band is just 18,400 km² (Barnes 2017). For comparison, the Arctic coastal region is ~6 million km² (Gattuso et al. 2006). The area is therefore very small. Barnes et al. (2020) estimate the carbon "sequestration" capacity of WAP fjords through generation of new zoobenthic biomass to be "> 780 tonnes C yr⁻¹" (the term "sequestration" within the Barnes papers and within the present manuscript is used very loosely; the IPCC defines the term differently, which is confusing). In any case, the ~800 t C yr⁻¹ of new zoobenthic biomass is basically insignificant compared to global CO₂ emissions

which are currently at 37 Gt C yr⁻¹. In comparison, global forests draw down 1.1 Gt C yr⁻¹ and macrophytes sequester up to 0.9 Gt C yr⁻¹ (Duarte 2017). In my view, for a carbon sink to be significant, it needs to act on the Gt scale.

There are two potential negative feedbacks to Antarctic Blue Carbon that are not really discussed. The first is benthic disturbance through iceberg grounding and scouring. It is persistent in these areas; every year it impacts a third of the seafloor at depths < 25 m, and for shallower waters it is > 90 % (Barnes 2017). This process will affect the proliferation of zoobenthic communities in newly exposed shallow areas and would disturb buried matter in the sediments. This point is mentioned briefly in the Introduction and Outlook sections (L53-57; L237-239) but its impact on Blue Carbon is not estimated. The second negative feedback is related to the erosion of marine-terminating glaciers that is required in order to expose new seabed. There is emerging evidence in the Arctic that marine-terminating glaciers support high productivity through rising subsurface meltwater plumes that transport large volumes of nutrient-rich deep waters to the surface, stimulating phytoplankton production (Meire et al. 2017). Fjords with only land-terminating glaciers lack this upwelling mechanism and have lower productivity. Since the carbon burial efficiency depends primarily on the supply of organic matter to the seafloor, it is not clear to what extent this negative feedback would offset Blue Carbon gains.

I struggled with the concept of autotrophic and heterotrophic benthic ecosystems being equated to carbon sinks and sources, respectively (L31-35 and elsewhere). I understand that there can be a net drawdown of CO₂ in autotrophic communities and vice-versa, but the fate of that carbon is ultimately what matters. Since this concept forms the foundations of the paper, there needs to be much clearer evidence that this is indeed the case.

In summary, I believe that this is a novel dataset that has potentially important ecological implications for our understanding of high-latitude systems. However, I believe the 'benthic carbon storage and sequestration' spin on this data should be largely eliminated as it raises numerous fundamental questions and works against what is otherwise a novel and interesting dataset.

Author reply:

Overall comments: We thank the reviewer very much for this very thorough and constructive review. We highly value the appreciation of our dataset and results and agree to adapt the framework of our study. We had originally placed our manuscript in a blue carbon framework, to raise awareness for the potential blue carbon function of future Antarctic seafloors (both shelf and deep sea), which is currently being investigated within the EU MSCA RISE project 'CoastCarb', joining 100+ Antarctic researchers, but the project has only recently started and the study is still in its infancy.

With respect to the quantification of two potential negative feedbacks on blue carbon: Ice scouring by growlers is studied in Potter Cove by one of our co-authors (Dolores Deregiibus), but not at the spatial resolution that we would need here. Similarly, there are burial estimates, based on ²¹⁰Pb sediment profiles (Monien et al. 2017), but not for each of our sampling sites. Since we currently cannot present quantitative estimates on the importance of blue carbon along the WAP, nor for the two potential negative feedbacks on blue carbon in Potter Cove specifically, we **agree with the reviewer to remove the blue carbon/carbon storage and sequestration framework** of the paper.

We also **omitted the statements on carbon sinks and sources (title, introduction)** and stayed with the more objective terms heterotrophy and autotrophy.

In addition, we **acknowledge the emerging evidence** that **marine-terminating glaciers support high productivity** through rising subsurface meltwater plumes (Meire et al. 2017), which was also recently observed along the WAP (Cape et al. 2019) (see reviewer comment 5). We also added PAR data to our manuscript (see reviewer comment 11). Though these PAR data are largely lacking for the sampling site Creek, they are informative to explain patterns in net community metabolism at Faro and Isla D.

References:

Monien D, Monien P, Brünjes R, Widmer T, Kappenberg A, Busso AA, Schnetger B, Brumsack HJ. Meltwater as a source of potentially bioavailable iron to Antarctica waters. *Antarctic Science*. 2017 Jun 1;29(3):277.

Meire L, Mortensen J, Meire P, Juul-Pedersen T, Sejr MK, Rysgaard S, Nygaard R, Huybrechts P, Meysman FJ. Marine-terminating glaciers sustain high productivity in Greenland fjords. *Global Change Biology*. 2017 Dec;23(12):5344-57.

Cape MR, Vernet M, Pettit EC, Wellner J, Truffer M, Akie G, Domack E, Leventer A, Smith CR, Huber BA. Circumpolar Deep Water impacts glacial meltwater export and coastal biogeochemical cycling along the west Antarctic Peninsula. *Frontiers in Marine Science*. 2019 Mar 26;6:144.

Detailed comments:

1. **Reviewer comment L32:** “Carbon sink” usually refers to carbon that is sequestered on geological timescales. Here and throughout the manuscript it would be good to align these terms (e.g. storage, sequestration, sink, burial, etc.) with the ‘official’ definitions by the IPCC, for clarity. L45-46: It would be good to include quantitative estimates- how big is the feedback, potentially? Is this locally/globally significant?

Author reply: As stated above, we decided to omit the carbon storage and sequestration framework and focus solely on benthic carbon balance (autotrophy/heterotrophy). The title has been reformulated as ‘*Glacial melt disturbance shifts the carbon balance of an Antarctic seafloor ecosystem to heterotrophy*’ and the corresponding paragraphs in abstract, introduction and discussion have been rephrased as:

Abstract:

L26-28: ‘*Climate change-induced glacial melt affects benthic ecosystems along the West Antarctic Peninsula, but current understanding of the effects on benthic primary production and respiration is limited.*’

L34-37: ‘*Ongoing accelerations in glacial melt and run-off may steer shallow Antarctic seafloor ecosystems towards net heterotrophy, altering the metabolic balance of benthic communities and potentially impacting the carbon balance and food webs at the Antarctic seafloor.*’

Introduction:

L42-46: ‘The West Antarctic Peninsula (WAP) has undergone rapid and significant warming during the second half of the 20th century¹. The sea ice season has shortened by about 100 days^{2,3} and 87% of coastal glaciers are in retreat^{4,5}. These alterations in the cryosphere have

strong consequences for marine ecosystems⁶, *but the effects on the metabolic balance of the benthic communities are poorly quantified.*

L55-60: *'As a result, more frequent ice scouring in coastal WAP areas may alter patterns in benthic community respiration and decomposition of dead material in the ice-scoured tracks. During the melt season sub-glacial melt induces yet another factor influencing the carbon balance in shallow systems, when an up to 5m-thick turbid water column layer develops.'*

L66-67: *'This shift can have important consequences for biogeochemical cycling, benthopelagic coupling and ultimately, food webs on the WAP.'*

Discussion:

L263-266: *'Given the evident fragility of these coastal Antarctic ecosystems already under climate change pressure, they should be protected against additional anthropogenic disturbance such as tourism to maintain their important role in the whole system's biogeochemistry and food web.'*

2. Reviewer comment L46: are in retreat

Author reply: OK. Corrected.

3. Reviewer comment L55: and consequently a

Author reply: OK. Corrected.

4. Reviewer comment L51-56: Here you describe a potential negative feedback, but you do not present it quantitatively within the context of Blue Carbon gains. Which is expected to be more important?

Author reply: Since we currently cannot quantify the potential losses and gains, as mentioned above, we decided to omit the blue carbon framework.

5. Reviewer comment L51-56: Marine-terminating glaciers support high productivity; their reduction could be another negative feedback. See work by Meire et al. 2017 in Global Change Biology.

Author reply: Agreed. We added the following sentence to the introduction L 50-53: *'However, marine-terminating glaciers may also support high productivity through rising subsurface meltwater plumes with nutrient-rich deep water^{12,13}. This effect disappears when these glaciers retreat on land¹².'*

References:

12. Meire, L. *et al.* Marine-terminating glaciers sustain high productivity in Greenland fjords. *Global Change Biology* **23**, 5344–5357 (2017).

13. Cape, M. R. *et al.* Circumpolar Deep Water Impacts Glacial Meltwater Export and Coastal Biogeochemical Cycling Along the West Antarctic Peninsula. *Front. Mar. Sci.* **6**, (2019).

6. **Reviewer comment** L64: There are multiple covariables that can affect the autotroph-heterotroph balance. Increased turbidity would decrease sunlight transmittance and increase sedimentation, both of which would shift the system towards net heterotrophy.

Author reply: Agreed. We added this to the following sentence: 'In shallow high latitude systems where benthic microalgae are the main primary producers, a reduction in photosynthetically active radiation availability *due to increased turbidity in combination with increased sedimentation*²² can determine a shift in benthic metabolism from an autotrophic to a heterotrophic state²³.'

Reference:

22. Hoffmann, R. et al. Implications of glacial melt-related processes on the potential primary production of a microphytobenthic community in Potter Cove (Antarctica). *Frontiers in Marine Science* 6, 655 (2019).

7. **Reviewer comment** L99: such as that seen

Author reply: OK. Corrected.

8. **Reviewer comment** L100: "...and then only before 2003." Please rephrase.

Author reply: OK. We rephrased the sentence as 'A long sea ice cover period such as that of 2015 *had been observed before 2003 and then again only in 2011*'.

9. **Reviewer comment** L143-144: "Apart from some patches at the least glacial-disturbed site Faro..." How was this assessed?

Author reply: Some benthic chambers at Faro in spring 2016 showed net production, while others did not. We refer to the new Figure 3a in which the NCM measurements of the individual chambers are shown as dots. Some of these NCM values are positive, meaning net production, while others are negative, meaning net respiration. This is most likely a result of patchy benthic microalgae distribution. This was evidently not clear in the previous version and has now been stated explicitly in the text: 'Apart from some patches at the least glacial-disturbed site Faro *in spring 2016*, there was no net benthic microalgal production in spring 2016 or winter and summer 2015 (*see some replicates with net production (positive NCM) in Figure 3a*).' (L155-157)

10. **Reviewer comment** L146-153: Negative GPP doesn't make intuitive sense- I guess this is a sensitivity limitation of the method. There are also many values close to zero- are they significantly different from zero?

Author reply: "Gross primary production [GPP] is then calculated as the sum of average NCM and the absolute value of average CR per site and sampling occasion." (Methodology, Line 475-477). GPP is not measured, but calculated as the sum of (positive) NCM and (negative) CR oxygen content in separate clear and dark chambers, respectively. For each treatment we have 3 replicates, which lead to one average (\pm sd) value for NCM and another one for CR. Then, one single GPP value is calculated as the sum of these averages per sampling occasion and so, we cannot statistically test a single GPP for any statistical difference with zero. In this sense, even intuitively, if respiration exceeds production, such as when organisms consume previously accumulated stock, negative GPP can be estimated. Similarly, close to zero values reflect the balance between both processes. We stated this now more explicitly in L477-479: 'Gross primary production [GPP] is then calculated as the sum of average *NCM (of triplicate*

transparent chambers) and the absolute value of average CR (of triplicate dark chambers) per site and sampling occasion. GPP can be negative when respiration exceeds production, such as when organisms consume previously accumulated stock. Similarly, close to zero values reflect the balance between both processes.'

11. **Reviewer comment** L140-153: Seabed light (PAR) data would be important here. Were any attempts made to constrain PAR? Did measurements fail?

Author reply: Yes, we made attempts to measure irradiance: in summer, winter and spring 2015, we used HOBO loggers, whereas in spring 2016, PAR measurements were made with PAR sensors, unfortunately without HOBO logger measurements in parallel. We had preferred not to present this data since they are not comparable in methodology and units. However, we decided to include data from long term PAR sensors that were deployed at Faro and Isla D at 10-11m water depth, a little deeper than the sites of benthic chambers deployment to answer to the reviewer's request. Supplementary Figure 1 now displays these data.

sensors (Odyssey Photosynthetic Irradiance Recording System, Data Flow Systems, Christchurch, New Zealand) installed approximately 0.5 m above the seafloor at approx. 10 m water depth at Faro and Isla D throughout 2015 and November-December 2016. Data for Isla D are only sparsely available since ice scouring frequently damaged the light sensors close to the glacier. Also at Faro, there was a lack of data in December 2015. The grey data points are from a location at ~500m distance from Faro at 11 m water depth. These PAR-sensors were calibrated according to Deregibus et al. (2016) and PAR was measured with a temporal resolution of 30 minutes. At Creek, PAR was measured only during the spring 2016 incubation, a Li-Cor PAR sensor (LI-192, Li-Cor Biosciences, Lincoln, Nebraska, USA; factory calibrated) with a temporal resolution of 1 s for 36 h. Red dots indicate the date of the in situ incubations and the experimentally determined light compensation point ($26 \mu\text{mol m}^{-2} \text{s}^{-1}$) (see methodology in Hoffmann et al. 2019).

Since PAR measurements lack in two out of three sampling occasions at Creek and in one occasion at Isla D, we could not take these data into account in the PCA or other analyses, but we do integrate the observed patterns in L151-155: ‘Net primary production by benthic microalgae (positive NCM) was mainly observed in spring 2015, when photosynthetic active radiation (PAR) measured at the seafloor clearly surpassed $26 \mu\text{mol m}^{-2} \text{s}^{-1}$, the experimentally determined light compensation point at Faro and Isla D²² (no PAR data for Creek) (Supplementary Figure 1).’ Similarly, in the paragraph in L155-167 we added: ‘Apart from some patches at the least glacial-disturbed site Faro in spring 2016, there was no net benthic microalgal production in spring 2016 or winter and summer 2015 (see some replicates with net production (positive NCM) in Figure 3a). This absence of net benthic production corresponds with PAR lower than the light compensation point²² in summer (Faro, Isla D) and winter 2015 (Faro), but not in spring 2016, when measured PAR at each site surpassed the light compensation point²² (Supplementary Figure 1). As discussed in Hoffmann et al.²², the low to absent benthic microalgal production in spring 2016 is probably not attributed to unavailability of PAR at the sea floor, but rather to direct physical disturbance of the benthic microalgae: damage of the benthic diatom photosynthetic apparatus can occur through sediment accumulation³⁷. Sedimentation also affects benthic microalgae through the longer distances that diatoms have to migrate to find the best light conditions³⁸, which affects energy allocation, thereby lowering the overall net production²².’

12. Reviewer comment L165: “...cluster clearly opposite...” Please rephrase.

Author reply: OK. We rephrased this as ‘In the PCA, these winter and Isla D samples cluster clearly opposite from the samples farther from the glacier (Faro and Creek) (Figure 4).’

13. Reviewer comment L175: “...better irradiance conditions...” You do not have data on this, so please state specifically how you can infer better conditions.

Author reply: This was indeed the case in the previous version. We now display irradiance data in Supplementary Figure 1. We further acknowledge that benthic metabolism is affected by both irradiance and sedimentation load. We specified this in the corresponding sentence: ‘Regression models clearly illustrate that net benthic metabolism (NCM) is enhanced by lower SPM concentrations in the water column (proxies for better irradiance conditions and/or lower sedimentation on the shallow water benthic microalgal assemblage) under low glacial run-off (Supplementary Table 4):...’ (L197-200).

14. Reviewer comment L176: “With” should have a small letter “w”

Author reply: OK. Corrected.

15. Reviewer comment L178: “...higher relative chl-a content...” Would you expect grazing to play a role here? In areas where grazing is intensive there can be a low standing stock but a high productivity due to high turnover of microalgal biomass.

Author reply: Yes, we also state this in L228-235: ‘An exception to the joint decline in benthic primary production and macrofauna biomass is the high recruitment of *L. elliptica* (cf. high densities, small individual biomass in Supplementary Figure 3, Figure 5) at the glacier front (Isla D) in spring 2016. This large burrowing bivalve is known to graze very efficiently on (resuspended) benthic diatoms⁴³. Hence, the very abundant local *L. elliptica* population

might have acted as a top-down control on the relatively low biomass of benthic microalgae close to the glacier front (Isla D) during spring 2016.'

- 16. Reviewer comment** L194: "...are also translated..." Please rephrase. Perhaps "...also result in...?"

Author reply: OK

- 17. Reviewer comment** L329-333: "Although meteorologically not correct..." Please rephrase and provide arguments as to why you select this approach.

Author reply: We added another argument to the argumentation we already provided to select this approach. The sentence was rephrased as: 'Although *strictly not in winter anymore*, for the sake of simplicity, we considered the sampling at Faro on 28 September 2015 as a winter sampling since it was performed only two weeks later than the winter sampling at Isla D *and the cove was still densely covered with sea ice*. Similarly, we still consider the sampling at Faro on 28 December 2015 as a spring sampling, since it was performed in the same expedition as the spring sampling at the other two sites.'

- 18. Reviewer comment** L353: "...after all expeditions except for winter 2015." You can simply say that the samples were either extracted immediately after collection or stored for later laboratory analysis. If you think that one approach biased the measurements, then this should be stated, but otherwise it is not required.

Author reply: We prefer to state explicitly that there can be a ~10% overestimation of absolute chl-*a* concentrations measured spectrophotometrically in winter 2015 (Lorenzen 1969) compared to the chl-*a* measurements with HPLC in the other seasons. This was already stated explicitly in L382-384.

- 19. Reviewer comment** L394: "overseen". Do you mean omitted?

Author reply: We meant literally not seen. We clarified this by adding some words to the sentence: 'Although diatoms can be identified to the species level, very large cells e.g. of *Gyrosigma* sp. can be *overlooked* with the applied method (e.g. *due to low abundance, lost during sample dilution*), hence not counted.' L415-418.

- 20. Reviewer comment** L467: "...remained stable during winter..." Is this a reasonable assumption to make?

Author reply: To justify this assumption we added a reference and the sentence: '*Both Arctic 78,79 and Antarctic studies 80 have shown that this is a reasonable assumption to make.*' L 496-497.

References:

78. Mazurkiewicz, M. *et al.* Seasonal constancy (summer vs. winter) of benthic size spectra in an Arctic fjord. *Polar Biol* **42**, 1255–1270 (2019).

79. Włodarska-Kowalczyk, M., Górska, B., Deja, K. & Morata, N. Do benthic meiofaunal and macrofaunal communities respond to seasonality in pelagial processes in an Arctic fjord (Kongsfjorden, Spitsbergen)? *Polar Biology* **39**, 2115–2129 (2016).

80. Glover, A. G., Smith, C. R., Mincks, S. L., Sumida, P. Y. G. & Thurber, A. R. Macrofaunal abundance and composition on the West Antarctic Peninsula continental shelf: Evidence for

a sediment 'food bank' and similarities to deep-sea habitats. *Deep Sea Research Part II: Topical Studies in Oceanography* **55**, 2491–2501 (2008).

21. **Reviewer comment** L470: "glmnet" What is this? Is it an R software package?

Author reply: Yes, the reference in the sentence refers to an R package. We stated this now more explicitly: 'To identify the best predicting parameters, a preselection of predictor parameters was performed using *the R package "glmnet"*⁸¹. L506.

Reviewer #2 (Remarks to the Author):

Brief summary

In their manuscript the authors combine metrological, glaciological and hydrological data with measurements of benthic metabolism in three stations of an Antarctic coastal bay, Potter Cove. They succeed in demonstrating that the overall climate change, portrayed in MEI and SAM indices, leading to periods of increased temperature, subsequent glacial melting and more runoff/water turbidity in the bay, may result in an overall shift from previously net phototrophic to future net heterotrophic metabolism in this system. The data presented for 2015/16 indeed are supportive of this change and it does fit the causal cascade of relations demonstrated with long term data, models and climatology. Since carbon sequestration by burial and transfer into the food chain is reduced, at least temporarily, under heterotrophic conditions, the well-founded conclusion is that this specific change in Potter Cove may portray the future development of several shallow coastal Antarctic marine environments.

Overall impression of the work.

My over impression is very positive and I enjoyed reading the manuscript. The text is written in a very compact way and readily understandable. (Drawing from my own limited experience with publishing in nature, the style is appropriate)

The specific data sets from 2015/16 from Potter Cove are a solid ground of data despite the missing macrozoobenthos data in winter 2015. The assumption used for macrozoobenthos, a stable population, is ok in order to perform the statistical analyses.

The strength of this manuscript comes from the combination with the ENSO Index and Southern Annular Mode indices, the observed melting and glacial retreat, turbidity and long monitoring data series at Potter Cove. Thus overall data represent different, but sufficiently long periods of observations in each case. Hydrographic data at the locations and finally benthic fauna and activity parameters relating to the three sites, Faro, Isla D and Creek, are merged well into a suited data set, that really only reaches its summed value with the research done at those three locations.

Despite the difficulties associated with it, glacial melt model, runoff in both creeks and groundwater flow seem to be captured well and sufficiently robust for the subsequent discussion on consequences for Potter Cove benthic systems.

The authors also dealt well with methodological differences (Chl-a spectrophotometric versus HPLC determination). Statistical testing is straight forward but correct, and augmented by a PCA analyses.

I am convinced that there is no (maybe non-published ...) data set that would allow to exercise an equivalent cascade of cause-effect relations in order to argue that the shift

towards reduced sequestration and thus the loss of a process abating CO₂ emissions is a likely scenario.

Author reply:

Overall comments: We thank the reviewer very much for his/her appreciation of our

work. **Specific comments:**

22. Reviewer comment Line 162: What does the word “these” relate to? Prior a gradient in sediment parameters is discussed, but “these” could either relate to far or distant from the glacier. Better write: was higher in Creek samples.

Author reply: We thank the reviewer for identifying this. We specified this in the corresponding sentence: ‘In addition, biomass of macrobenthos excluding the large burrowing bivalve *Laternula elliptica* was higher in the samples of Faro and Creek than at Isla D (Supplementary Figure 2).’

23. Reviewer comment Line 166: Spell Tab instead of Table

Author reply: We checked the formatting in Communications Biology and it seems ‘Table’ should be spelled ‘Table’.

24. Reviewer comment Line 167: shouldn’t this be Fig. S1 only?

Author reply: The sentence refers to renumbered Supplementary Figure 2 (all biomass including meiofauna) and renumbered Supplementary Figure 3 (*Laternula* biomass). We have changed this accordingly in the text.

25. Reviewer comment Line 201/202 AND 205/207: This first sentence seems much too general to be true. If net respiration surmounts primary production, there may still be as much food for grazers or predators. Heterotrophy by bacteria may support the CO₂-efflux, increased activity by otherwise constant macro-biomass may still persist. So at least on a short time scale this statement cannot be kept at such a general level. It should be specified. Also the statement is repeated in 205/207. This is repetitive and should be avoided.

Author reply: We agree that this sentence was too general. We rephrased this as ‘If benthic net carbon respiration *remains* higher than primary production *on longer time scales*, a reduction in the available biomass for benthic consumers might be expected’ and omitted the repetitive sentence a few lines further down.

26. Reviewer comment Line 240: “into the atmosphere through respiration”. Yes there would be an increased return to the atmosphere. But rather decomposition of dead material than respiration.

Author reply: Agreed. Sentence has been changed to: ‘These processes would lead to an increased return of CO₂ into the atmosphere through *decomposition of dead material*.’

27. Reviewer comment Line 313: The sentence starting “An important and ...” does not seem to continue correctly, particularly the “and” somehow does not seem to fit the message.

Author reply: Thank you for pointing this out. We split the sentence in two as ‘*An important and difficult task is to define the capture zone of subglacial and glacial run-off. The Potter Creek river basins were defined based on the topography and drainage networks survey.*’

28. **Reviewer comment** Line 425-128: I am missing the explicit information as to which individuals of *Laternula* were weighed for the siphon width-AFDW relation either missing or not expressed sufficiently obvious.

Author reply: The method to derive the conversion factor to estimate AFDW of *Laternula elliptica* from siphon width was published in Hoffmann et al. (2018). We agree that the described methodology in our manuscript might be confusing, so we reformulated the sentence as ‘Assuming a linear relationship between siphon width and AFDW, *biomass of L. elliptica* was estimated from the siphon width – AFDW conversion factor calculated in Hoffmann et al.³⁹’

Reference:

39. Hoffmann, R. *et al.* Spatial variability of biogeochemistry in shallow coastal benthic communities of Potter Cove (Antarctica) and the impact of a melting glacier. *PLoS one* **13**, e0207917 (2018).

Reviewer #3 (Remarks to the Author):

This manuscript is very interesting. It combines climatology, glaciology, oceanography, and benthic biogeochemistry to construct a narrative about how an ecosystem responds to different climatic forces (an El Niño cold winter vs a La Niña warm winter) which offers insight to how climate change could impact the ecosystem’s function in terms of carbon storage. The manuscript is comprehensive but not over-long. I think it is worthy of publication in *Communications Biology*. I have listed a few comments below that would help the manuscript before potential publication.

Reviewer comment My biggest concern with the entire manuscript is the term “blue carbon”. It is immediately used in the text, but never circled back to until the very last sentence of the Outlook section. Blue carbon is such a buzz word, and I would suggest that the manuscript is just as strong without invoking this term and simply referring to “carbon sources” and “carbon sinks”, as is done in the title. It is usually used in the context of salt marsh, mangroves, and seagrasses, but I see by the reference titles that the Antarctic coastal ecosystems have been referred to as possessing blue carbon despite no vegetative ecosystem that can bury recalcitrant organic matter. Can you elaborate on the usage of this term for the WAP? It appears some of the references (particularly the Barnes publications) have previously used this term. Is there ample sedimentation that occurs to bury C before remineralization? Can you provide a range of sedimentation/burial rates that occur for the area (cm yr⁻¹) and carbon sequestration rates (g C m⁻² yr⁻¹)? Can you comment on the mechanisms that preserve algal C against decomposition processes so that burial of C does occur? I would argue that the “blue carbon” term is used so that ecosystem services of threatened coastal habitats could be monetized so their protection is incentivized, but perhaps this is a myopic view of the term. It is not to say that any marine ecosystem cannot sequester/bury/store carbon for durations that could potentially interrupt the carbon-climate change feedback loops; indeed studying how an ecosystem is a sink or source of carbon is extremely relevant. Again, I suggest the authors take out this word from the few instances where it occurs. But if they feel strongly about including this study with the

other blue carbon literature, then perhaps they could justify its use by briefly addressing some of the above questions for readers like me.

Author reply: We thank the reviewer very much for the very constructive review. After reading the comments of reviewers 1 and 3 and as already mentioned in our answer to reviewer 1, we decided to omit the blue carbon (carbon storage and sequestration) framework and rephrase the context to importance for benthic biogeochemistry, benthic-pelagic coupling and foodwebs. We currently do not have the necessary data to support our statements on carbon storage and sequestration, but we are strongly involved in efforts to quantify this in the near future.

Specific comments:

29. Reviewer comment L43-44: too many commas. "...host productive benthic communities, which in combination with a relatively high burial rate in fjord systems contributes..."

Author reply: This sentence has been removed in accordance with Reviewer comment 1.

30. Reviewer comment Fig 2. Make axis text bigger please. It would be helpful to have season labels, perhaps at the top of the graph columns, that denote where "summer 2015", "winter 2015", "spring 2015", and "spring 2016" occur. It is a little confusing when looking between text to figure since spring occurs almost on top of the New Year.

Author reply: The axis text has been enlarged. We present the different incubation seasons with different colours. The legend states: 'vertical dashed lines indicate the dates of the in situ incubations *in summer 2015 (orange), winter 2015 (blue), spring 2015 (green) and spring 2016 (grey)*'. A colour-blind-friendly colour palette was used.

Figure 2. Environmental parameters. **a**, Monthly averages of the multivariate ENSO index (MEI v2) and Southern Annular Mode (SAM) **b**, average monthly air temperature **c**, daily sea ice presence **d**, monthly averaged modelled glacial discharge **e**, wind speed and direction

during the months of the *in situ* incubations, with contours indicating the percentage of time the wind had a particular direction **f**, speed average sea surface temperature in the surface waters (0–1 m water depth) of inner Potter Cove (filled dots) **g**, bottom water temperature measured during benthic chamber incubations at each site (open triangles) **h**, SPM and **i**, chl-*a* in the surface waters (0–1 m water depth) of inner Potter Cove from 2014–2017. In panels a-d and f-i, grey ribbons represent standard deviation; vertical dashed lines indicate the dates of the *in situ* incubations *in summer 2015 (orange), winter 2015 (blue), spring 2015 (green) and spring 2016 (grey)*; grey zones indicate the astronomical austral winter; limited data coverage of sea surface temperature and salinity in autumn and winter 2015 and 2016 is shown as disconnected data points.

31. Reviewer comment L174-181: Since the study doesn't measure irradiance/PAR directly, it is not exactly correct to invoke it here. Could you instead say something like "the measured parameters that are proxies for PAR/irradiance attenuation"?

Author reply: We agree with this and, as mentioned in our answer to Reviewer 1 comment 11, have added PAR data from slightly deeper waters (10-11 m water depth instead of 6-9 m) in Supplementary Figure 1, which we also integrate in the text (see answer to Reviewer comment 11). However, since these data are incomplete, we cannot use them in the regression models. Therefore, we integrated the reviewer's suggestions in the text: 'Regression models clearly illustrate that net benthic metabolism (NCM) is enhanced by *lower SPM concentrations in the water column (proxies for better irradiance conditions or lower sedimentation load on the shallow water benthic microalgal assemblage) under low glacial run-off (Supplementary Table 4)*'

32. Reviewer comment Table S4: "SSSalinity" should be defined somewhere. Unless it is a typo?

Author reply: the Supplementary Table has been updated and does not contain SSSalinity in the model. The correct one is the following one:

Supplementary Table 4: Significant models resulting from regression analysis. Models only include significant ($p < 0.05$) partial regression coefficients. *NCM* = Net Community Metabolism; *SPM_w* = Suspended Particulate Matter in the water column. *chl-a:CPE* = chl-a:Chloroplast Pigment Equivalent ratio; *MPB biomass* = Microphytobenthos (benthic diatom) biomass

	Model	N	R ² _{adj}	p-value
NCM	NCM = -35 -1.26 x SPM _w + 0.73 x chl-a:CPE	30	0.67	< 0.001
chl-a:CPE	chl-a:CPE = 47+ 0.18 x MPB biomass	37	0.32	< 0.001

We also made sure that the abbreviations in the other tables are spelled out in the captions.

33. Reviewer comment L208: “at at”

Author reply: OK

34. Reviewer comment L106: add a qualifier like “until later that summer...” regarding the glacial water discharge since a few months later it was quite high.

Author reply: OK. The text has been added: ‘As such, cold air temperatures during early spring 2015 prevented glacier melting and high particle-laden freshwater to enter Potter Cove waters *until later that summer*’.

35. Reviewer comment L018-111: The data support this claim; however, Dec 2016-Jan 2017 is also had equally high chlorophyll concentrations. Can you comment on this occurrence since the 2016 winter was so different but still resulted in high water column chlorophyll?

Author reply: We thank the reviewer for this comment. We now comment on this phenomenon at the end of the narrative on the La Niña year (L142-146): ‘*The high, short lived chl-a values detected in Jan 2017 were probably advected from neighbouring ice-free and particle-free areas by the dominating westerly winds. This allowed phytoplankton to grow during a short period (3 d) of relatively calm winds, but they were swept away when wind intensity increased again, as on January 16, 2017.*’

36. Reviewer comment L129-133: The austral summer of Dec 2016-Jan 2017 appears to have the same magnitude discharge as the austral summer of Dec 2015-Jan 2016. The text explanation confused me claiming that the discharge in Dec 2016-Jan 2017 was 3x higher than any preceding summer. But clearly the summer of Dec 2016-Jan 2017 was approximately equal. Please clarify the text here.

Author reply: We agree that this can be confusing. We stated more explicitly that we are comparing max. run-off numbers for each austral summer: ‘The peak in glacial run-off during austral summer 2015/2016 occurred in February 2016, but maximum run-off was ~10 m³ s⁻¹ (Figure 2d), about three times higher than in the years 2010–2015³⁰, (L 117-119). ... ‘the subsequent austral summer of 2016 – 2017 was characterized by exceptionally high (max. in February 2017 ~14 m³ s⁻¹, about 4.5 times higher than the 2010 – 2015 maxima) glacier discharge rates (Figure 2d), ...’ (L132-136).

REVIEWERS' COMMENTS:

Reviewer #1 (Remarks to the Author):

I commend the authors for their thorough response. I think this paper will influence thinking in the field and will stimulate research on the links between the benthic compartment and climatic processes. I only have a couple of further points which the authors may wish to consider.

L1: Title. I am glad you removed reference to carbon sinks and carbon sources. I would, however, advise against mentioning 'carbon' in the title, simply because you measure the oxygen flux and not the carbon flux. The flux measurements are described as 'metabolism' in the methods; I think this is more appropriate.

L146-153 in the original document: Negative GPP. I appreciate the response to my comment, and I agree that, in practice, negative GPP can result due to the way it is calculated. We should remember, however, as users of this method that GPP holds a real-world significance, i.e. it is the photosynthetic rate. While you may end up with a negative number due to the method used, GPP cannot, by definition, be negative. The lowest it can ever be is zero. It is more correct to say, in the methods, that any values below zero were just assumed to be zero. It is however a rather small point and I will leave it up to the authors to decide this.

Reviewer #3 (Remarks to the Author):

I am satisfied with how the authors addressed my previous comments.